# Redefining the role of AMPK in autophagy and the energy stress response

Ji-Man Park ◎[1], Da-Hye Lee[1] & Do-Hyung Kim ◎[1,2,3,4] ✉

Autophagy maintains cellular homeostasis during low energy states. According to the current understanding, glucose-depleted cells induce autophagy through AMPK, the primary energy-sensing kinase, to acquire energy for survival. However, contrary to the prevailing concept, our study demonstrates that AMPK inhibits ULK1, the kinase responsible for autophagy initiation, thereby suppressing autophagy. We found that glucose starvation suppresses amino acid starvation-induced stimulation of ULK1-Atg14-Vps34 signaling via AMPK activation. During an energy crisis caused by mitochondrial dysfunction, the LKB1-AMPK axis inhibits ULK1 activation and autophagy induction, even under amino acid starvation. Despite its inhibitory effect, AMPK protects the ULK1-associated autophagy machinery from caspase-mediated degradation during energy deficiency, preserving the cellular ability to initiate autophagy and restore homeostasis once the stress subsides. Our findings reveal that dual functions of AMPK, restraining abrupt induction of autophagy upon energy shortage while preserving essential autophagy components, are crucial to maintain cellular homeostasis and survival during energy stress.

Glucose starvation poses a challenge to the survival of eukaryotic cells by causing energy crisis. Macroautophagy (hereafter referred to as autophagy) is widely recognized as a mechanism that can provide energy to glucose-starved cells for survival. However, the precise role of autophagy in this process has been controversial[1–3]. While the energy-providing role of autophagy is often emphasized, it is important to note that autophagy itself requires energy for membrane rearrangement, trafficking, and completion[4–7]. In glucose-starved cells, autophagy induction may depend on the ability of the cells to obtain a minimum level of energy required for the process, which can be derived from stored energy reserves, such as lipids or glycogen, or other substrates. When this ability is limited, cells may prioritize the use of available energy for other vital processes rather than autophagy. From this energetic perspective, it is unclear whether autophagy is the primary cellular response to energy stress.

Over the past decade, research in the field has largely relied on the concept that glucose starvation induces autophagy through the activation of 5′-adenosine monophosphate-activated protein kinase (AMPK). According to the prevailing model, AMPK phosphorylates and activates ULK1 (UNC-51 like kinase 1) to induce autophagy[8,9]. Mechanistic target of rapamycin complex 1 (mTORC1), the key negative regulator of autophagy, prevents the function of AMPK by disrupting the interaction between AMPK and ULK1. However, the model could not explain several recent reports. Inhibition of mTORC1 reduced AMPK-mediated phosphorylation of ULK1 Ser556 (mouse Ser555)[10,11], the phosphorylation considered to promote autophagy[9]. Nutrient starvation diminished, rather than enhanced, the binding between AMPK and ULK1[12,13]. A769662, an allosteric activator of AMPK, suppressed autophagosome formation[11]. AICAR (aminoimidazole-4-carboxamide riboside) and metformin, which induce AMPK activation, inhibited or failed to induce autophagy[14–16], and AMPK knockdown increased autophagy in pancreatic β cells[17]. These reports have raised questions about the validity of the predominant concept.

Although the prevailing concept regarding the positive role of AMPK in autophagy was reported more than a decade ago[8,9], the role of AMPK in regulating ULK1 within cells has never been thoroughly

[1]Department of Biochemistry, Molecular Biology and Biophysics, University of Minnesota, Minneapolis, MN 55455, USA. [2]Institute for Diabetes, Obesity and Metabolism, University of Minnesota, Minneapolis, MN 55455, USA. [3]Center for Immunology, University of Minnesota, Minneapolis, MN 55455, USA. [4]Masonic Cancer Center, University of Minnesota, Minneapolis, MN 55455, USA. ✉e-mail: dhkim@umn.edu

validated. In the past decade, several cellular substrates of ULK1 have been identified, enabling researchers to measure the specific activity of ULK1 within cells[18–27]. Using those tools, we have analyzed how AMPK regulates ULK1 activity and found that AMPK inhibits, rather than promotes, ULK1 activity and autophagy induction. We have determined that two AMPK-mediated phosphorylations of ULK1 are crucial for the inhibition of ULK1 activity and the suppression of autophagy induction.

## Results

### AMPK suppresses ULK1 signaling to the autophagy initiation machinery

According to the widely accepted concept in the field, AMPK induces autophagy by phosphorylating ULK1 Ser556 (mouse Ser555)[9]. However, we recently showed that the phosphorylation is suppressed by Torin1[10], the drug that inhibits mTOR and induces autophagy. We confirmed the result using a different cell line (Fig. 1a). The phosphorylation completely disappeared in amino acid-starved cells. Rapamycin also reduced the phosphorylation, albeit to a lesser extent. The dependence of Ser556 phosphorylation on mTORC1 activity resembles the response of ULK1 Ser758 (mouse ULK1 Ser757) phosphorylation to mTORC1 activity[8]. Moreover, we found that mTORC1 inhibition does not stabilize but disrupts the interaction between AMPK and ULK1 (Fig. 1b). This result of interaction is consistent with the report by Shang et al.[12], but disagrees with the prevailing model proposed by Kim et al.[8]. Our finding that ULK1 is dissociated from AMPK in mTORC1-inhibited cells provides a more plausible explanation for how mTORC1 inhibition reduces AMPK-mediated phosphorylation of ULK1 Ser556.

After observing the inverse relationship between AMPK-dependent phosphorylation of ULK1 at Ser556 and mTOR inhibitory conditions that induce autophagy, we asked how AMPK affects ULK1 activity within cells. Unexpectedly, we found that A769662 blocked ULK1-mediated phosphorylations of Atg14 and Beclin 1, which are key autophagy regulators[21–23], in response to amino acid starvation in mouse embryonic fibroblasts (MEFs) and HCT116 cells (Fig. 1c–e and Supplementary Fig. 1a, b). A769662 also suppressed the phosphorylations of other ULK1 targets, such as Atg16L1 and Atg13. The suppression occurred as early as 10 minutes after AMPK activation, which is reflected by pULK1 Ser556 and pACC Ser79 (Fig. 1c). Moreover, A769662 suppressed Torin1-induced phosphorylations of Atg14 and Beclin 1 (Fig. 1d, e and Supplementary Fig. 1c, d). We further confirmed the inhibitory effect of A769662 on ULK1 activation in multiple cell lines and mouse primary hepatocytes (Supplementary Fig. 1e–i). Other AMPK stimulating agents, such as 991, MK8722, and GSK621, also showed similar suppressive effects in cells treated with amino acid starvation or Torin1 (Fig. 1f, g and Supplementary Fig. 1i–k). Throughout all the results, we consistently observed an inverse correlation between ULK1 activity and its phosphorylation at Ser556 (Fig. 1c–g).

To further clarify how AMPK regulates ULK1, we analyzed ULK1 activity in cells lacking AMPK. Depletion of AMPKα1 and α2 (AMPKα DKO) increased ULK1-mediated phosphorylations of Atg14 and Beclin 1 (Fig. 1h, i and Supplementary Fig. 1l). A769662 was unable to inhibit ULK1 activity in AMPK-deficient cells (Fig. 1j, k and Supplementary Fig. 1m, n). Reconstitution of wild-type (WT) AMPKα1 or α2, but not its kinase-dead (KD) mutant, restored the suppressive effect of A769662 (Fig. 1l and Supplementary Fig. 1o, p), confirming that the suppression of ULK1 activity by A769662 is through AMPK activation. We also confirmed that AMPK activation suppresses Torin1-induced activation of ULK1 in the mouse liver and skeletal muscle (Fig. 1m, n). Overall, these results demonstrate that AMPK negatively regulates ULK1 signaling to the autophagy initiation machinery (Fig. 1o).

Compound C, an AMPK inhibitor previously used to validate the positive role of AMPK on ULK1 activity[8], suppressed ULK1 activity even

in the absence of AMPK (Supplementary Fig. 1q, r), indicating that the drug is not a reliable tool to study ULK1 activity. We attempted to demonstrate the inhibitory effect of AMPK on ULK1 activity in vitro, but the results were inconclusive. The incubation of ULK1 with recombinant AMPKα/β/γ complex moderately reduced the kinase activity of ULK1 toward phosphorylation of Atg14 Ser29 (Supplementary Fig. 1s). ULK1 was able to autophosphorylate Ser556 in the absence of AMPK. This suggests that the in vitro assay may not appropriately reflect AMPK-specific effects on ULK1 activity.

### Glucose starvation and phenformin suppress ULK1 activity

Glucose starvation for 2 h, which enhanced AMPK activity, almost completely suppressed amino acid starvation-induced activation of ULK1 in HCT116 cells and MEFs (Fig. 2a, b and Supplementary Fig. 2a). The suppression did not occur in AMPK-deficient cells. Glucose starvation also suppressed Torin1-induced activation of ULK1 in various cell types, including adipocytes, myoblasts, hepatoma cells, and mouse primary hepatocytes (Fig. 2c and Supplementary Fig. 2b). The ULK1 activity was gradually reduced as glucose levels decreased and AMPK activity increased in mouse primary hepatocytes (Fig. 2d, e). The glucose starvation effect was mimicked by 2-deoxy-D-glucose (2-DG), which depletes cellular ATP and stimulates AMPK activity (Supplementary Fig. 2c).

Phenformin, a drug that reduces cellular ATP levels and activates AMPK, suppressed Torin1- or amino acid starvation-induced activation of ULK1 (Fig. 2f, g and Supplementary Figs. 1i and 2d). The suppression was marginal in cells lacking AMPKα or liver kinase B1 (LKB1), the protein kinase that activates AMPK in response to increases in the cellular AMP to ATP ratio (Fig. 2f–i and Supplementary Fig. 2e, f). Consistent with this, phenformin did not suppress the activity of ULK1 in HeLa cells that naturally lack LKB1 (Supplementary Fig. 2g). The suppression was restored upon reconstitution of WT LKB1 in HeLa cells, but the effect was much weaker upon reconstitution of KD LKB1 (Fig. 2j). Combined, these results suggest that the suppression of ULK1 activity by phenformin is mediated through the LKB1-AMPK pathway.

### Mitochondrial inhibitors suppress ULK1 activity through the LKB1-AMPK pathway

We wondered if mitochondrial inhibitors, which increase AMPK activity, can suppress ULK1 activity. We tested oligomycin A, antimycin A, rotenone, and carbonyl cyanide 3-chlorophenylhydrazone (CCCP), and found that they all reduce ULK1 activity (Fig. 2k and Supplementary Fig. 2h). Oligomycin A showed a minimal inhibitory effect after 1 hour, but a stronger effect after 6 h. CCCP exhibited the weakest effect probably due to its robust inhibition of mTORC1. Even in cells lacking AMPK, CCCP was able to inhibit mTORC1 through an unknown mechanism (Fig. 2k). The mitochondrial inhibitors were unable to suppress ULK1 activity in AMPK-deficient cells (Fig. 2k), demonstrating that the inhibition was via AMPK. The suppressive effects became greater when cells were subjected to either amino acid starvation or Torin1 treatment (Fig. 2l and Supplementary Fig. 2i–k). The suppressive effects were not observed in MEFs and HCT116 cells lacking AMPK or LKB1 (Fig. 2l, m), and in HeLa and A549 cells that naturally lack LKB1 (Supplementary Fig. 2l). These findings demonstrate that the functional mitochondria are necessary for ULK1 activation by mTORC1 inhibition and that the LKB1-AMPK axis mediates the mitochondrial effect (Fig. 2n).

The results above consistently demonstrate that AMPK inhibits ULK1 activity in cells under conditions of energy stress or mitochondrial dysfunction. This inhibitory effect of AMPK activation is dominant over the stimulatory effect of mTORC1 inhibition. We observed that AMPK activation drastically reduced S6K1 Thr389 phosphorylation, but it had a relatively marginal effect on ULK1 Ser758 phosphorylation (Figs. 1a, d, f, 2k and Supplementary Fig. 2h–j). This suggests that AMPK may primarily regulate autophagy through direct

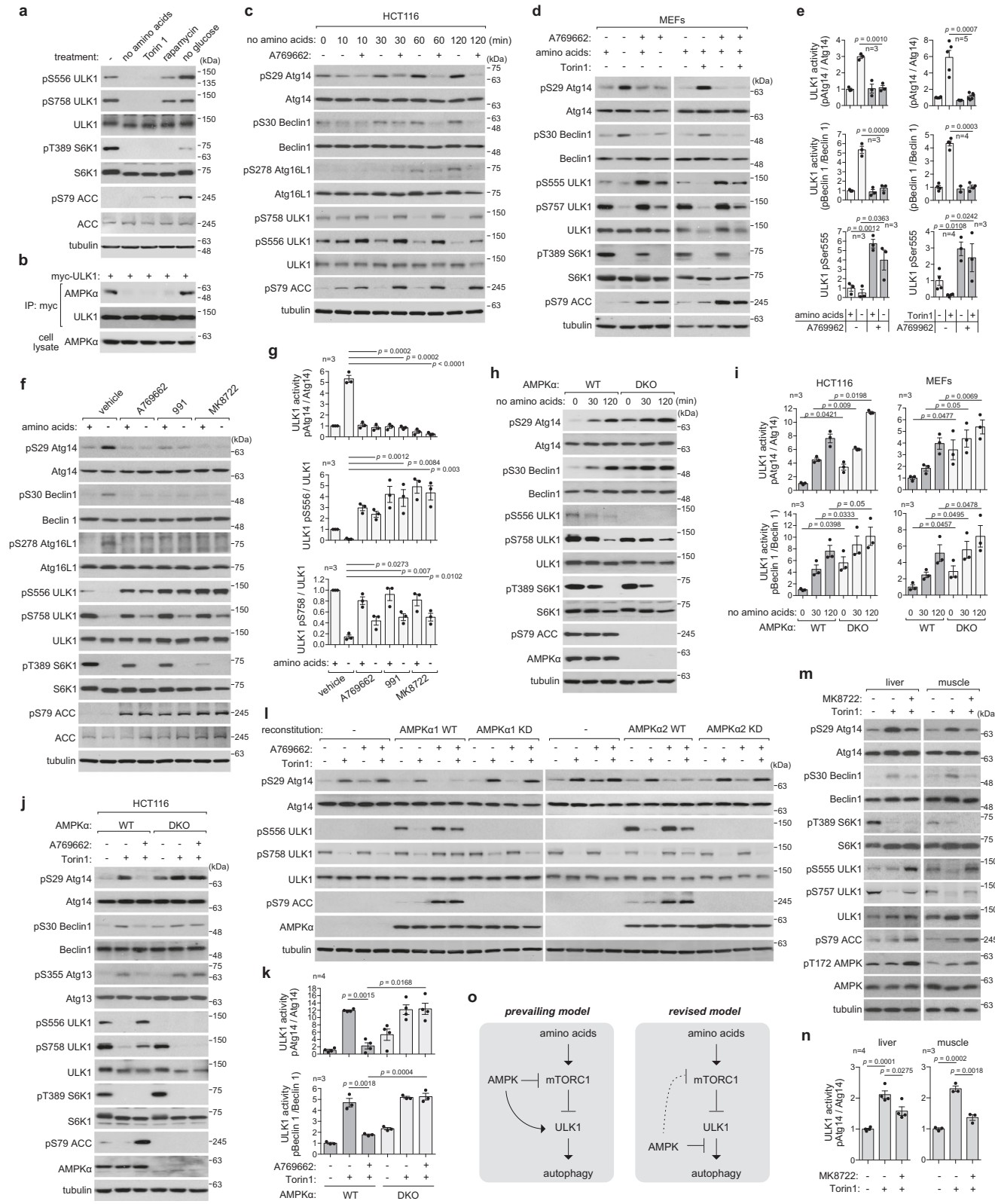

phosphorylation of ULK1 (Fig. 1o) rather than through inhibiting mTORC1[28,29].

## AMPK inhibits amino acid starvation-induced autophagy, but it is necessary for optimal autophagy

ULK1 phosphorylates and activates the Atg14-associated Vps34 complex to induce the formation of phagophores and

autophagosomes[19,21–23]. We wondered whether AMPK activation can suppress the Vps34 activity. Glucose starvation or A769662 reduced the activation of the Vps34 complex in response to amino acid starvation (Fig. 3a and Supplementary Fig. 3a, b). The suppression was significantly compromised or not detected for the Vps34 complex from AMPK-deficient cells. Despite the suppressive effect, the removal of AMPK reduced the Vps34 activity, suggesting that AMPK is

**Fig. 1 | AMPK inhibits ULK1 signaling to the autophagy initiation machinery.**
**a** mTORC1 inhibition suppresses ULK1 Ser556 phosphorylation. HCT116 cells were starved of amino acids or glucose for 2 h or treated with Torin1 (250 nM) or rapamycin (100 nM) for 1 h. **b** mTORC1 inhibition disrupts the AMPK-ULK1 interaction. ULK1 KO HEK293T cells transiently transduced with myc-ULK1 were treated as in **a**. Anti-myc immunoprecipitates were analyzed for endogenous AMPKα. **c** AMPK activation suppresses ULK1 activity. HCT116 cells were starved of amino acids in the presence or absence of A769662 (100 μM). **d, e** A769662 prevents the ability of amino acid starvation or Torin1 to increase ULK1 activity. MEFs were pre-treated with vehicle (−) or A769662 (+) for 30 min, then incubated with either full or amino acid-deprived medium or treated with Torin1 for 1 h. **f, g** AMPK activators suppress ULK1 activity. HCT116 cells were treated with A769662, 991 (10 μM) or MK8722 (10 μM) for 30 min then starved of amino acids for 2 h. **h, i** AMPKα depletion enhances ULK1 activity. AMPKα WT and DKO HCT116 cells were starved of amino

acids during the indicated periods. **j, k** AMPKα depletion blunts the suppressive effect of A769662 on ULK1 activity. HCT116 cells were treated with A769662 (100 μM) and/or Torin1 (250 nM) for 1 h. **l** AMPK kinase activity is necessary to suppress ULK1 activity. AMPKα DKO HEK293T (left) and HCT116 cells (right) stably reconstituted with WT or KD AMPKα were treated as described in **j**. **m, n** AMPK activation suppresses Torin1-induced ULK1 activation in the liver and skeletal muscle (tibialis anterior). C57BL6 mouse was intraperitoneally injected with Torin1 alone or together with MK8722 for 1 h. **o** Model for AMPK role in autophagy. The dotted line indicates that the AMPK-mTORC1 axis plays a lesser role in autophagy. Data in the graphs of this figure are shown as mean ± standard error of the mean (SEM). Data were analyzed by two-tailed Student's t-test. The number of independent experiments is denoted by the '*n*' value on each graph. Source data are provided as a Source Data file.

necessary for optimal Vps34 activation. We speculated that this might be due to direct phosphorylation of Beclin 1 Ser93 (mouse Ser91) by AMPK[30]. However, the Beclin 1 phosphorylation did not correlate with Vps34 activity (Fig. 3a and Supplementary Fig. 3a, c). The Beclin 1 phosphorylation still occurred in AMPK-deficient cells, indicating that the phosphorylation needs further characterization.

Consistent with the suppression of Vps34 activity, A769662 decreased amino acid starvation-induced formation of WIPI2 and LC3B puncta, indicative of phagophore and autophagosome formation, respectively (Fig. 3b, c and Supplementary Fig. 3d, e). This finding is consistent with a recent report showing that A769662 suppressed amino acid starvation-induced formation of autophagosomes[11]. A769662 also suppressed autophagy flux during amino acid starvation in MEFs (Fig. 3d, e and Supplementary Fig. 3f, g) and in HCT116 cells (Supplementary Fig. 3h–l). The suppression was blunted in AMPK-deficient cells, suggesting that the drug effect was mediated by AMPK. Despite the suppressive effect, AMPK depletion significantly reduced amino acid starvation-induced phagophore and autophagosome formation as well as autophagy flux (Fig. 3b–e and Supplementary Fig. 3d–l).

Treatment with phenformin, antimycin A, and rotenone also reduced amino acid starvation-induced formation of WIPI2 and LC3B puncta (Fig. 3f, g) and suppressed autophagy flux (Fig. 3h, i and Supplementary Fig. 3m). Similarly, 2-DG also suppressed autophagy flux triggered by amino acid starvation (Supplementary Fig. 3n, o). Glucose starvation impaired autophagy flux in cells fully supplied with amino acids (Supplementary Fig. 3p–r). After 21 h without glucose, autophagy flux was almost completely blocked. In the similar way, A769662 inhibited basal autophagy flux in cells grown in a complete medium. We also demonstrated that MK8722, an AMPK-activating agent whose efficacy was validated in vivo[31], suppressed the formation of LC3B puncta and the increase in the level of LC3B II induced by Torin1 in the liver and skeletal muscle (Fig. 3j–m and Supplementary Fig. 3s). Collectively, these results demonstrate that AMPK activation suppresses autophagy induction, but its presence is necessary for optimal Vps34 activation and autophagy (Fig. 3n).

### AMPK activation impairs the removal of protein aggregates via autophagy
After observing that AMPK activation suppresses autophagy, we wondered whether AMPK negatively regulates the clearance of protein aggregates through the selective autophagy called aggrephagy[32,33]. An established method for assessing aggrephagy is exposing cells to puromycin, a drug that prematurely terminates translation leading to the formation of defective ribosomal products and a buildup of ubiquitinated protein aggregates[34]. Treatment with puromycin for 4 hours resulted in the formation of protein aggregates that comprised both ubiquitinated proteins and p62 (Fig. 3o, p and Supplementary Fig. 3t). Upon removal of puromycin, protein aggregates were cleared through autophagy (Supplementary Fig. 3u). AMPK activation via

chemical activators (A769662, 991, and MK8722), glucose starvation, and 2-DG treatment suppressed the aggregate clearance in MEFs (Fig. 3p, q and Supplementary Fig. 3v) and HeLa cells (Supplementary Fig. 3w, x). The inhibition did not occur in AMPK-deficient cells. Phenformin and metformin suppressed aggrephagy in MEFs, and the effect was not observed in MEFs lacking AMPK or in HeLa cells naturally deficient in LKB1. This suggests that the suppression of aggrephagy was mediated through the LKB1-AMPK axis. We further confirmed the suppressive effect using HT22 neuronal cells (Supplementary Fig. 3y, z).

### AMPK stably binds to ULK1 when mTORC1 phosphorylates ULK1 Ser758
To understand the mechanism underlying the inhibitory role of AMPK in autophagy, we hypothesized that the inhibitory effect of AMPK might depend on its interaction with ULK1. We showed in Fig. 1b that mTORC1 inhibition disrupts the AMPK-ULK1 interaction. Replacing ULK1 Ser758 with alanine (S758A) impaired the AMPK-ULK1 interaction (Fig. 4a and Supplementary Fig. 4a, b), which is consistent with the observations by Kim et al.[8] and Shang et al.[12]. ULK1 Ser638 is regulated by both mTORC1 and AMPK for its phosphorylation[12], but its substitution of alanine (S638A) did not alter the AMPK-ULK1 interaction (Supplementary Fig. 4b), indicating that Ser758 is the major mTORC1 target site that regulates the AMPK-ULK1 interaction. To examine the effect of mTORC1 activation on the interaction, we tested overexpression of Rheb (Ras homolog enriched in brain), a small GTPase that activates mTORC1. Rheb overexpression enhanced Ser758 phosphorylation and stabilized the AMPK-ULK1 interaction (Fig. 4a). The stabilization effect did not occur with either S758A or the S757C mouse ULK1, which was designed to mimic the dephosphorylated state of the residue[8]. This result demonstrates that mTORC1-mediated phosphorylation of Ser758 stabilizes the AMPK-ULK1 interaction, contradicting the previously established model[8]. The reason for the discrepancy is uncertain. In our study, we examined the interaction with endogenous AMPKα, rather than overexpressing AMPKα, to avoid any potential artifacts that could arise from non-physiological stoichiometric interaction with endogenous AMPKβ and AMPKγ.

### AMPK activation stabilizes its interaction with ULK1
As the increased interaction between AMPK and ULK1 correlated with ULK1 inhibition, we hypothesized that AMPK activation might enhance the interaction. Supporting the hypothesis, treatment with A769662 stabilized the interaction and prevented its disruption caused by amino acid starvation or Torin1 (Fig. 4b, c). The stabilization required the kinase activity of AMPK (Fig. 4d). The AMPK-ULK1 interaction was also stabilized by other AMPK activators (991, MK8722, GSK621), as well as by glucose starvation, 2-DG, phenformin (Fig. 4e and Supplementary Fig. 4c, d), and mitochondrial inhibitors such as antimycin A and rotenone (Fig. 4f). The interaction was also stabilized by CCCP albeit to a lesser extent. This could be attributed to the strong

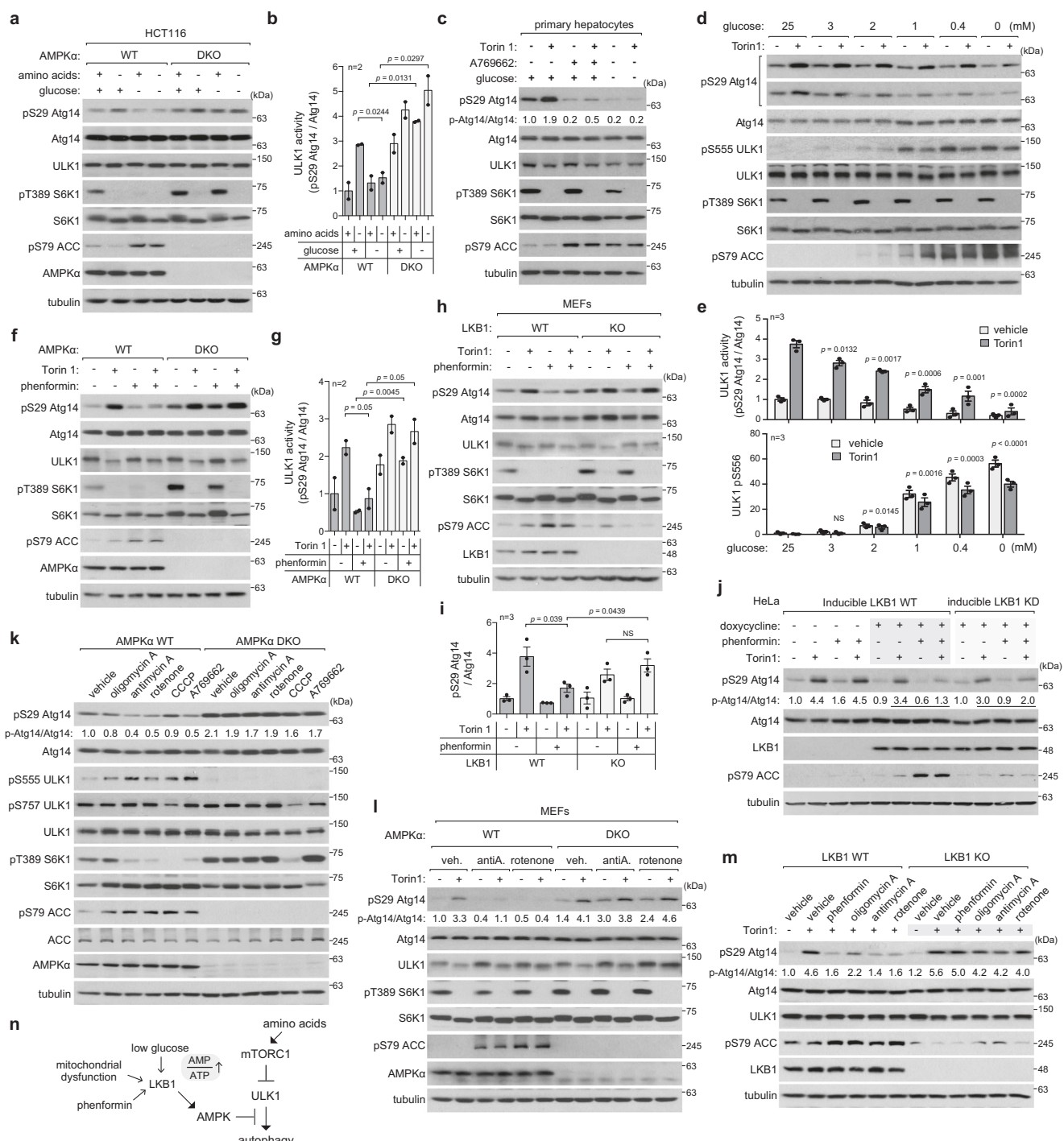

**Fig. 2 | LKB1-AMPK axis inhibits ULK1-autophagy signaling during energy stress. a**, **b** Glucose starvation suppresses amino acid starvation-induced ULK1 activation. Cells were starved of glucose for 2 h, then incubated in either full or amino acid-deprived medium for 2 h. Glucose condition remained unchanged during amino acid starvation. **c** Glucose starvation and A769662 suppress ULK1 activity in primary hepatocytes. Mouse primary hepatocytes were treated with A769662 (100 µM) or starved of glucose for 2 h, followed by incubation with either a vehicle (-) or Torin1 (250 nM) for 1 h. **d**, **e** Decreasing glucose levels in culture reduces ULK1 activity. Mouse primary hepatocytes were incubated in medium containing varying concentrations of glucose for 2 h, followed by treatment with a vehicle or Torin1 (250 nM) for 1 h. The *p* values are relative to Torin1-treated cells cultured at 25 mM glucose (NS, not significant). **f**, **g** Phenformin suppresses Torin1-induced ULK1 activation via AMPK. HCT116 cells were treated with phenformin (2 mM) for 2 h, followed by treatment with Torin1 for 1 h. **h**, **i** Suppression of ULK1

activity by phenformin requires LKB1. Cells were treated as in **f**. **j** Reconstitution of WT LKB1, but not KD mutant, in HeLa cells restored the inhibitory effect of phenformin on ULK1 activity. HeLa cells were reconstituted with inducible WT or KD mutant LKB1 and treated with doxycycline for 16 h for LKB1 expression. The cells were treated as in **f**. **k** Mitochondrial inhibitors suppress ULK1 activity through AMPK. MEFs were treated with oligomycin A (10 µM), antimycin A (10 µM), rotenone (1 µM), CCCP (25 µM), or A769662 for 1 h. **l** Mitochondrial inhibitors suppress Torin1-induced ULK1 activation through AMPK. MEFs were treated with mitochondrial inhibitors for 30 min, then with a vehicle (−) or Torin1 (+) for 1 h. **m** Inhibitory effects of mitochondrial drugs on ULK1 activity depends on LKB1. LKB1 WT and KO MEFs were treated as described in **k**, **l**. **n** Schematics for how LKB1-AMPK axis regulates autophagy. The statistical analysis in this figure was performed as in Fig. 1. Source data are provided as a Source Data file.

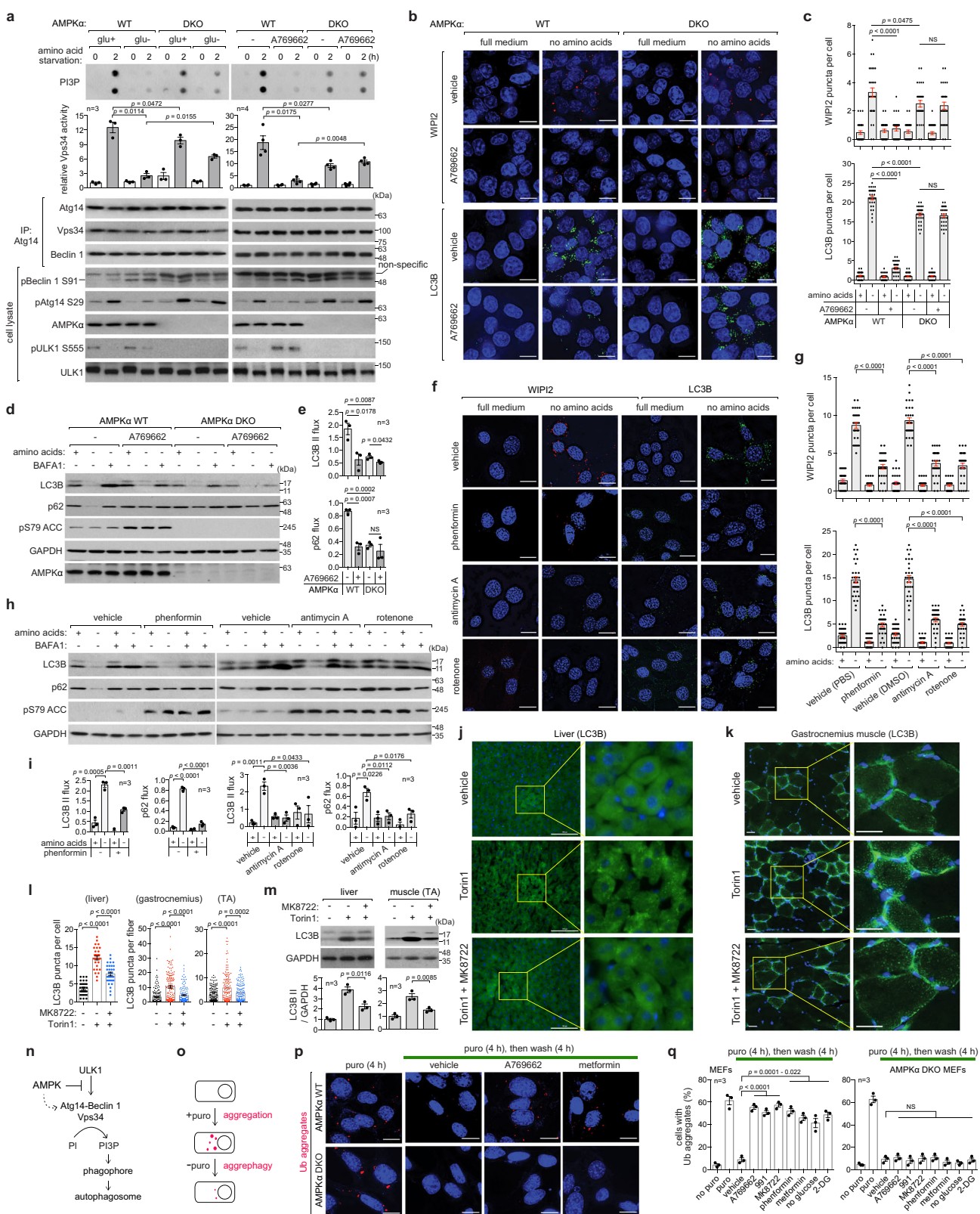

inhibitory effect of the drug on mTORC1 (Fig. 2k and Supplementary Fig. 2h–j). Collectively, these results demonstrate that AMPK activation protects the AMPK-ULK1 interaction against disruption by amino acid starvation or mTORC1 inhibition (Fig. 4g). Previously, Shang et al.[12] showed that AMPK has a strong binding affinity for ULK1 in full medium. However, the mechanism by which AMPK regulates the interaction was not explored in that study. Our finding indicates that when

activated, AMPK strongly binds to ULK1. As a result, AMPK blocks the activation of ULK1 and the initiation of autophagy even under amino acid starvation.

AMPK activation preserved the phosphorylation of ULK1 Ser758 under amino acid starvation (Fig. 4d and Supplementary Fig. 4e). This effect was not observed in cells lacking AMPK and in AMPKα DKO cells reconstituted with the KD mutant of AMPKα1. This finding suggests

**Fig. 3 | AMPK inhibits autophagy induction but supports optimal autophagy.**
**a** AMPK activation suppresses Atg14-Vps34 lipid kinase activity. The indicated MEFs
were starved of glucose or treated with A769662, as described in Fig. 2. Anti-Atg14
immunoprecipitates were isolated to measure Vps34 activity of producing PI-3-
phosphate (PI3P). **b, c** AMPK activation suppresses phagophore and autophago-
some formation during amino acid starvation. The indicated HCT116 cells were
starved of amino acids for 2 h in the presence or absence of A769662. WIPI2 and
LC3B puncta were analyzed in each cell. A total of more than 30 cells were analyzed
across three independent experiments. **d, e** AMPK activation suppresses autophagy
flux during amino acid starvation. The indicated MEFs were starved of amino acids
for 2 h in the presence or absence of A769662 and BAFA1 (200 nM). **f, g** Phenformin
and mitochondrial inhibitors suppress autophagosome formation during amino
acid starvation. MEFs were starved of amino acids in the presence of the chemicals
for 2 h. WIPI2 and LC3B puncta were analyzed as described in **b, c. h, i** Phenformin

and mitochondrial inhibitors suppress autophagy flux. MEFs were starved of amino
acids in the presence of the chemicals for 4 h. **j–m** MK8722 inhibits Torin1-induced
autophagy in the mouse liver and gastrocnemius and tibialis anterior (TA) skeletal
muscle. The mouse was treated as described in Fig. 1m. LC3B puncta stained using
anti-LC3B antibody were analyzed in liver cells and muscle fibers. A total of more
than 30 liver cells and 200 muscle fibers were analyzed using three animals for each
condition. **n** AMPK inhibits ULK1 signaling but supports optimal activation of Vps34
complex. **o** Schematics of aggrephagy assay. **p, q** AMPK activation suppresses
aggrephagy in MEFs. The y-axis in **q** represents the percentage of cells that contain
aggregates. Each analysis was performed on 50 cells across three independent
experiments. The statistical analysis in this figure was performed as in Fig. 1. In the
cell images, nuclei were stained with DAPI (blue). Scale bars indicate 10, 25, and 100
μm for cells, muscle, and liver, respectively. Source data are provided as a Source
Data file.

that AMPK activation prevents the loss of Ser758 phosphorylation
thereby stabilizing the interaction between AMPK and ULK1. The sta-
bilization effect by AMPK did not depend on either ULK1 kinase activity
or its binding to Atg8 (Supplementary Fig. 4f). The S758A mutation
almost completely abolished the AMPK-ULK1 interaction, but a low
level of interaction was still detected (Fig. 4h). The remaining inter-
action was destabilized by amino acid starvation and stabilized by
A769662. ULK1 mutants with S638A, S638D, S758D, or S638D/S758D
mutations showed similar interaction with AMPK as WT ULK1 in
response to amino acid starvation and A769662 (Supplementary
Fig. 4g, h). These results suggest that AMPK-dependent events, beyond
mTORC1-mediated phosphorylation of Ser758, stabilize the AMPK-
ULK1 interaction.

### AMPK stabilizes its binding with ULK1 by phosphorylating ULK1 Ser556 and Thr660

Previous studies using mass spectrometry have identified several resi-
dues in ULK1 as targets for AMPK phosphorylation, which include
murine ULK1 residues Ser317, Ser467, Ser555 (human ULK1 Ser556),
Thr574 (human Thr575), Ser637 (human Ser638), and Ser777[8,9] (Fig. 4i).
All these residues are conserved in human ULK1, except for Ser777.
Replacing these residues with alanine, either individually or in combi-
nation, did not affect the AMPK-ULK1 interaction or the ability of
A769662 to stabilize the interaction (Fig. 4j and Supplementary Fig. 4i,
j). We also evaluated Ser495 and Thr660 (murine Ser494 and Thr659),
two potential AMPK target sites identified through mass spectrometry[13].
Interestingly, the combination of alanine mutations at Ser495 and
Thr660 with S467A/S556A/T575A mutations resulted in the loss of the
stabilizing effect of A769662 on the interaction (Fig. 4j). Analysis of
various combinations of mutations revealed that the S556A and T660A
(AA) mutations are minimally required to abolish the ability of
A769662 to stabilize the interaction (Fig. 4k, l). We developed poly-
clonal antibodies specific for Thr660 phosphorylation and confirmed
that AMPK is responsible for the phosphorylation of the residue
(Supplementary Fig. 4k, l). The single mutation of either Ser556 or
Thr660 was not sufficient to abolish the ability of A769662 to stabilize
the AMPK-ULK1 interaction, and their substitution with aspartate or
glutamate did not lead to stabilization of the interaction without
A769662 (Supplementary Fig. 4m). These results suggest that the
phosphorylation of Ser556 or Thr660 is critical in maintaining the
stability of the AMPK-ULK1 interaction, but additional phosphoryla-
tions play a role in reinforcing the stability.

### AMPK inhibits ULK1 activity by phosphorylating ULK1 Ser556 and Thr660

Given that AMPK phosphorylates ULK1 Ser556 and Thr660 to stabilize
the AMPK-ULK1 interaction, we investigated the potential inhibitory
effects of those phosphorylations on ULK1 activity. We stably expres-
sed WT or AA ULK1 in HCT116 cells and MEFs where endogenous ULK1
and ULK2 were depleted, and analyzed ULK1 activity.

A769662 suppressed ULK1 activity toward the phosphorylations of
Atg14 and Beclin 1 under amino acid starvation or Torin1 treatment in
WT ULK1-reconstituted cells but not in AA ULK1-reconstituted cells
(Fig. 5a–c). Similarly, glucose starvation suppressed ULK1 activity in
WT cells during amino acid starvation, but the suppression was sig-
nificantly compromised in AA cells (Fig. 5d, e). To further validate the
inhibitory role of the phosphorylations, we tested doxycycline-
induced WT and AA ULK1 expressed at endogenous ULK1 levels.
Similarly to the results above, A769662 effectively suppressed WT
ULK1 activity but had a minimal effect on AA ULK1 activity (Fig. 5f, g
and Supplementary Fig. 5a). A769662 sustained Ser758 phosphoryla-
tion under amino acid starvation in WT cells but not in AA cells (Fig. 5a,
f). These results suggest that AMPK-mediated phosphorylations at
Ser556 and Thr660 stabilize Ser758 phosphorylation, enhancing the
stability of the AMPK-ULK1 interaction and suppressing ULK1 activa-
tion (Fig. 5h). Even in the presence of A769662, the S758A mutation
greatly enhanced ULK1 activity, whether it was combined with S638A
or not (Fig. 4h and Supplementary Fig. 5b), supporting the inhibitory
role of Ser556 and Thr660 phosphorylations that is dependent on
Ser758 phosphorylation.

Replacing either Ser556 or thr660 with alanine individually did
not abolish the ability of A769662 to preserve Ser758 phosphorylation
and suppress ULK1 activity (Supplementary Fig. 5c). The ability of
A769662 to suppress ULK1 activity was maintained by all mutants
except those that harbor both S556A and T660A mutations (Supple-
mentary Fig. 5d–f). The 4SA mutation, previously shown to suppress
mitophagy[9], moderately increased ULK1 activity under Torin1 treat-
ment instead of reducing it. We also tested the effect of alanine
replacement for Atg13 Ser224, a potential AMPK target site[35], on ULK1
activity without observing any significant effect (Supplementary
Fig. 5g). Atg9a, which is recruited to ULK1 in response to AMPK
activation[13], was also not necessary for the inhibitory effect of AMPK on
ULK1 activity (Supplementary Fig. 5h).

### AMPK inhibits autophagy by phosphorylating ULK1 Ser556 and Thr660

Next, we asked how Ser556 and Thr660 phosphorylations affect
autophagy. Glucose starvation suppressed amino acid starvation-
induced activation of the Atg14-Vps34 complex to a lesser extent in AA
cells compared to WT cells (Fig. 6a). Similarly, the inhibitory effect of
A769662 on Vps34 activity was blunted in AA cells compared to
WT cells (Fig. 6a and Supplementary Fig. 6a, b). However, the AA
mutation could not completely blunt the suppressive effect of AMPK
activation on Vps34 activity, suggesting that other AMPK-mediated
events contribute to the suppression of Vps34 activity.

The AA mutation had no effect on the formation of phagophores
and autophagosomes in response to amino acid starvation in A769662-
untreated cells (Fig. 6b, c and Supplementary Fig. 6c, d). Upon treat-
ment with A769662, the formation of phagophores and autophago-
somes was significantly reduced in WT cells, while the reduction was

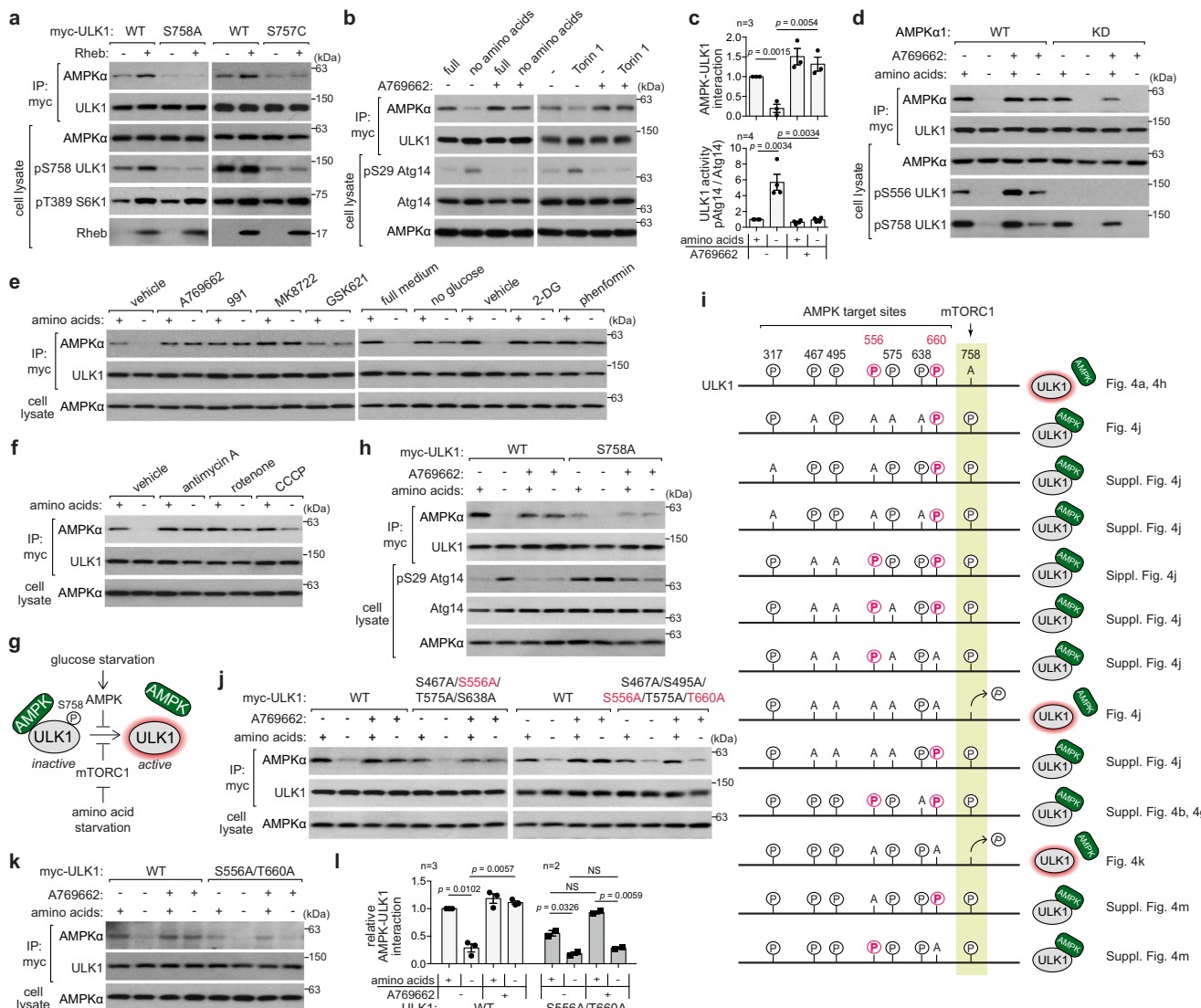

**Fig. 4 | AMPK stabilizes its binding with ULK1 by phosphorylating ULK1 Ser556 and Thr660. a** The phosphorylation of ULK1 at Ser758 by mTORC1 stabilizes the AMPK-ULK1 interaction. HEK293T cells lacking ULK1 were transiently transfected with either myc-tagged WT or mutant ULK1, with or without Rheb. Anti-myc immunoprecipitates were analyzed for the presence of endogenous AMPKα. **b** AMPK activation stabilizes the AMPK-ULK1 interaction. HEK293T cells lacking ULK1 were transiently transfected with myc-tagged ULK1. The cells were incubated in either full or amino acid-deprived medium, or treatment with Torin1 in the presence or absence of A769662 for 2 h. Anti-myc immunoprecipitates were analyzed for endogenous AMPKα. **c** Quantitative analysis of the AMPK-ULK1 interaction and ULK1 activity from **b**. **d** AMPK kinase activity is required to stabilize the AMPK-ULK1 interaction. HEK293T cells lacking AMPKα were transiently transfected with either WT or KD AMPKα1 and myc-tagged ULK1. The cells were treated as described in **b**. **e, f** AMPK activation stabilizes the AMPK-ULK1 interaction. ULK1 KO HEK293T cells that were transduced with myc-ULK1 were treated with the indicated

conditions for 1 h. **g** Schematics summarizing how mTORC1 and AMPK regulate the AMPK-ULK1 interaction and ULK1 activity. **h** AMPK-dependent events, beyond the phosphorylation of ULK1 Ser758 by mTORC1, stabilize the AMPK-ULK1 interaction. ULK1 KO HEK293T cells were transiently transfected with myc-tagged WT or the S758A ULK1 mutant. The cells were treated as described in **b**. **i** Summary of AMPK- and mTORC1-mediated phosphorylation sites on ULK1 and the impact of their alanine substitution on Ser758 phosphorylation and the stability of the AMPK-ULK1 interaction. The residue numbers are based on the human ULK1 sequence. **j, k** The phosphorylation of ULK1 at either Ser556 or Thr660 is necessary for AMPK to stabilize the AMPK-ULK1 interaction. **l** Quantitative analysis of the AMPK-ULK1 interaction from **k**. The statistical analysis in this figure was performed as described in Fig. 1. The concentrations of the chemicals used for the experiments in this figure are the same as described in Figs. 1, 2 and Supplementary Fig. 2. Source data are provided as a Source Data file.

significantly blunted in AA cells. The suppressive effect of A769662 on autophagy flux was also significantly blunted in AA MEFs (Fig. 6d, e and Supplementary Fig. 6e) and in AA HCT116 cells (Supplementary Fig. 6f, g). Similarly, glucose starvation suppressed amino acid starvation-induced autophagy flux in WT cells but barely in AA HCT116 cells (Fig. 6f, g and Supplementary Fig. 6h). These results suggest that AMPK activation suppresses amino acid starvation-induced increases in the activity of the Vps34 complex and autophagy through ULK1 phosphorylations at Ser556 and Thr660 (Fig. 6h). We also found that the AA mutation significantly reduced the suppressive effect of A769662 on

aggrephagy (Fig. 6i, j). However, a significant level of suppression was still observed with aggrephagy in AA cells treated with A769662, suggesting that AMPK activation suppresses aggrephagy through not only inhibition of ULK1 but also through additional unknown mechanisms.

## AMPK enables cells to maintain autophagy capacity and extends cell survival during glucose starvation by phosphorylating ULK1 Ser556 and Thr660

Our findings may seem counterintuitive, as both AMPK and autophagy play critical roles in cell survival[36–39]. We could confirm the critical role

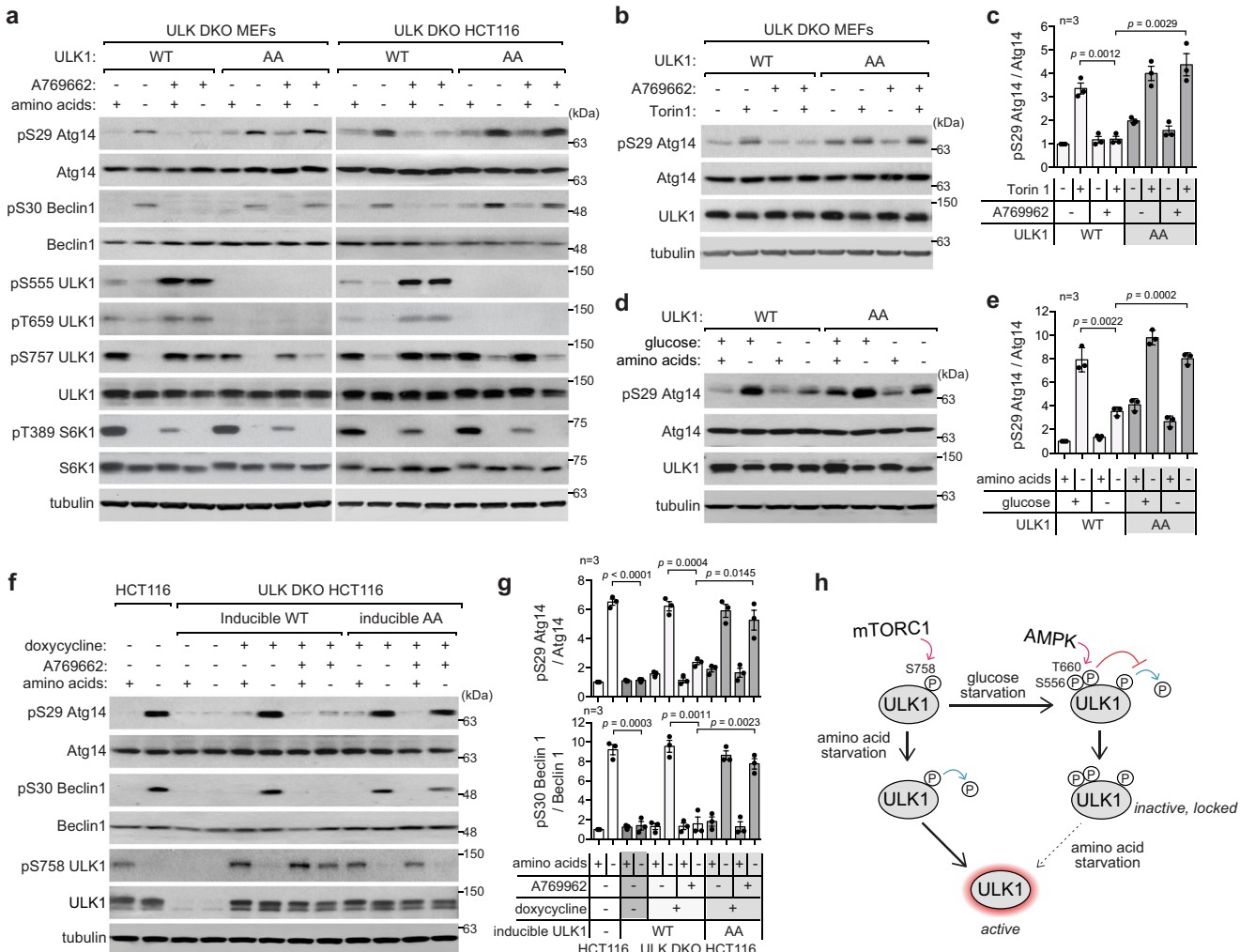

**Fig. 5 | AMPK inhibits ULK1 activity by phosphorylating ULK1 Ser556 and Thr660. a** The phosphorylation of ULK1 at Ser556 and Thr660 by AMPK suppresses ULK1 activity. ULK DKO MEFs and HCT116 cells were stably transduced with either WT or S556A/T660A (AA) ULK1 construct. The cells were incubated with either full or amino acid-deprived medium in the presence or absence of A769662 (100 μM) for 1 h. **b, c** The phosphorylation of mouse ULK1 at Ser555 or Thr659 by AMPK suppresses ULK1 activity. ULK DKO MEFs were stably transduced with either the mouse WT or the S555A/T659A (AA) mutant ULK1. The cells were treated with A769662 (100 μM) for 30 min, then subjected to further treatment with either the vehicle (−) or Torin1 (+) at a concentration of 250 nM for 1 h. **d, e** The phosphorylation of ULK1 at Ser556 and Thr660 by AMPK is necessary for glucose starvation to suppress ULK1 activity. ULK DKO HCT116 cells were stably transduced with either

the WT or the AA ULK1 construct. The cells were treated as described in Fig. 2a. **f** The inhibitory role of the phosphorylation of ULK1 at Ser556 and Thr660 was confirmed through use of an inducible form of ULK1. ULK DKO HCT116 cells were stably transduced with either the WT or the AA construct, whose expression was controlled by a doxycycline-dependent inducible promoter. The transduced cells were treated with doxycycline (100 ng/ml) for 16 h to induce the expression of ULK1 to levels comparable to that of the endogenous ULK1. The cells were then treated as described in **a**. **g** Quantitative analysis of ULK1 activity from **f**. **h** Diagram summarizing the roles of the phosphorylations of ULK1 by mTORC1 and AMPK in regulating ULK1 activity. The statistical analysis in this figure was performed as described in Fig. 1. Source data are provided as a Source Data file.

of the LKB1-AMPK-ULK1 pathway in cell survival during glucose starvation or phenformin treatment using HCT116 cells and MEFs that lacked those genes (Fig. 7a and Supplementary Fig. 7a–c). We wondered how this result can be reconciled with the negative role of AMPK in autophagy. We considered the possibility that AMPK plays the vital role in cell survival during glucose starvation not by inducing autophagy but by suppressing it. The ULK1 AA mutation is a suitable tool to test this hypothesis, as it allows us to disrupt the AMPK-autophagy link without perturbing other AMPK-mediated functions. HCT116 and MEF cells reconstituted with the AA mutant ULK1 showed a significant increase in apoptosis following prolonged glucose starvation, compared to cells reconstituted with the WT ULK1 (Fig. 7a, b and Supplementary Fig. 7a, b). Phenformin also induced apoptosis to a much greater extent in AA cells than in WT cells (Supplementary Fig. 7c). Prolonged amino acid starvation also induced apoptosis in HCT116 cells, which was suppressed by A769662 in WT ULK1-reconstituted

cells, but not in AA ULK1-reconstituted cells (Fig. 7c). These results suggest that AMPK-mediated phosphorylations of Ser556 and Thr660 protect glucose- or amino acid-starved cells against apoptosis.

During the prolonged glucose or amino acid starvation, ULK1 levels were drastically decreased (Fig. 7a, c and Supplementary Fig. 7a). The decrease was not attributed to changes in gene expression, autophagy, or the proteasome but was preventable by inhibiting caspase activity (Fig. 7d and Supplementary Fig. 7d, e). AMPK deficiency exacerbated the decrease in ULK1 levels, whereas treatment with A769662 showed a protective effect against the decrease in ULK1 levels (Fig. 7a, d–f and Supplementary Fig. 7a, c, f–j). Using cycloheximide, we confirmed that the decrease of ULK1 levels during glucose starvation is a result of protein degradation (Fig. 7g, h). In contrast to the WT ULK1, the AA mutant ULK1 was less stable during prolonged glucose starvation or when treated with phenformin (Fig. 7a, d–f and Supplementary Fig. 7a, c). AMPK was also required to protect ULK1-

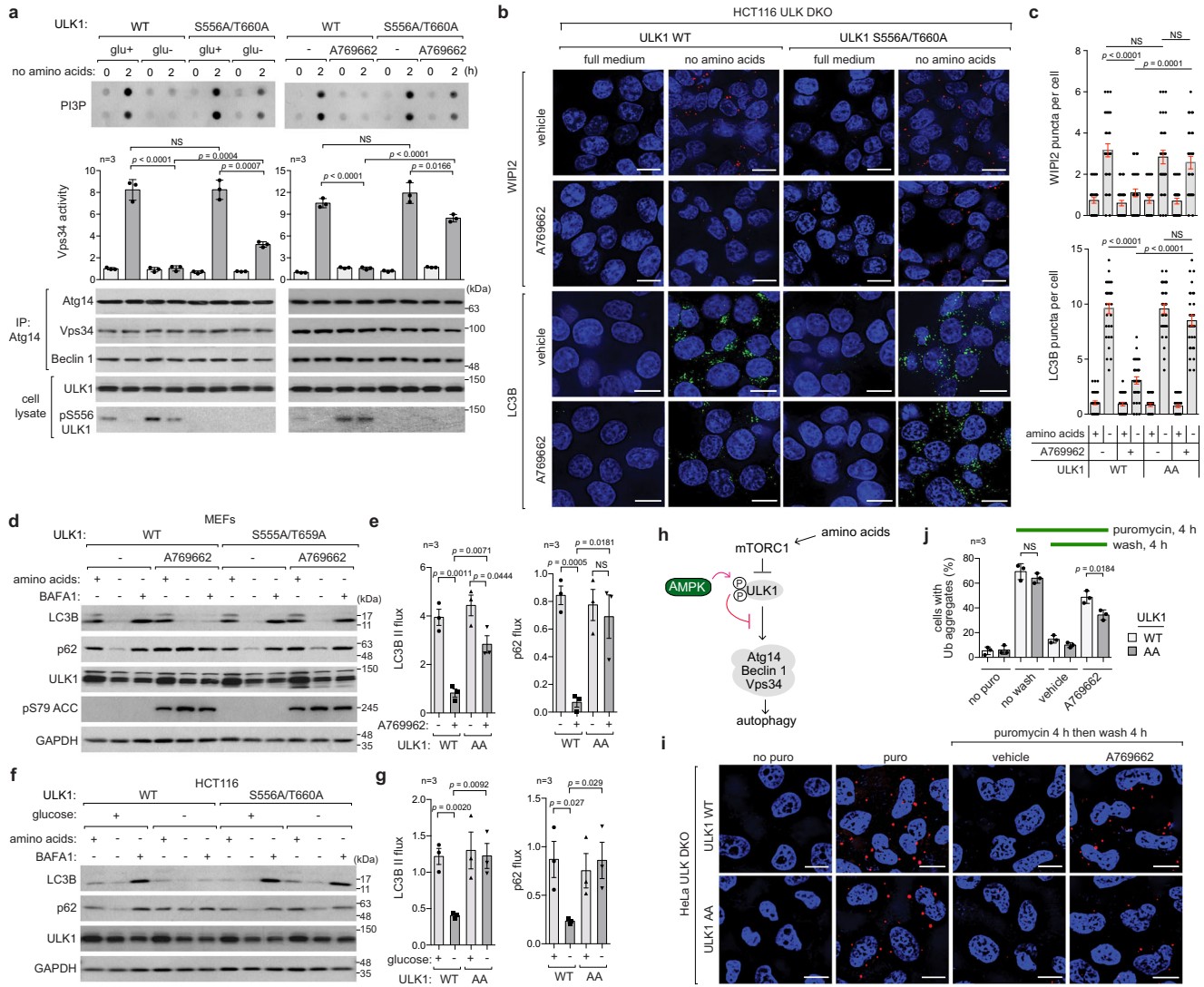

**Fig. 6 | AMPK suppresses Atg14-associated Vps34 activity and autophagy by phosphorylating ULK1 Ser556 and Thr660. a** AMPK suppresses Atg14-associated Vps34 activity through the phosphorylation of ULK1 at Ser556 and Thr660. ULK DKO HCT116 cells reconstituted with either the WT or the AA ULK1 were treated as described in Fig. 3a. Anti-Atg14 immunoprecipitates were isolated to measure Vps34 activity. **b**, **c** AMPK suppresses the formation of autophagosomes through the phosphorylation of ULK1 at Ser556 and Thr660 during amino acid starvation. The WT and AA ULK1-reconstituted cells, which were prepared as described in **a**, were subjected to the treatment conditions indicated in Fig. 3b. **d**, **e** AMPK suppresses autophagy flux through the phosphorylation of ULK1 at Ser556 and Thr660. ULK DKO MEFs reconstituted with either WT or AA ULK1 were starved of amino acids for 4 h in the presence or absence of A769662 and BAFA1. **f**, **g** Glucose

starvation-induced suppression of autophagy flux depends on ULK1 phosphorylation at Ser556 and Thr660 by AMPK. WT and AA HCT116 cells were pre-starved of glucose for 2 h, then analyzed for autophagy flux after a 4 h incubation period. **h** Diagram showing how AMPK negatively regulates ULK1 signaling to the Atg14-Vps34 complex. **i** AMPK phosphorylation of ULK1 Ser556 and Thr660 suppresses aggrephagy. ULK DKO HeLa cells reconstituted with either WT or AA ULK1 were treated with puromycin for 4 h, then incubated in full medium with or without A769662 (100 μM) for an additional 4 h. Ubiquitin-positive protein aggregates were analyzed by immunostaining (red). **j** Quantitative analysis of **i**. In the cell images, the nuclei were stained with DAPI (blue). A scale bar of 10 μm was used. The statistical analysis in this figure was performed as described in Figs. 1 and 3. Source data are provided as a Source Data file.

associated autophagy regulators, such as FIP200, Atg13, Atg14, Beclin 1, and Vps34, against degradation during glucose starvation (Fig. 7d, g, h and Supplementary Fig. 7k–n). Collectively, these results suggest that AMPK protects ULK1 and its associated autophagy machinery from degradation during glucose starvation.

Given that AMPK stabilizes the autophagy machinery components, we hypothesized that AMPK-deficient cells might lose their ability to initiate autophagy following prolonged glucose starvation. As expected, cells without AMPK showed a significant reduction in ULK1 levels and were impaired in the ability to activate ULK1 after 24 hours of glucose starvation (Fig. 7i–k). Similarly, cells reconstituted with the AA mutant ULK1 showed drastic reductions in both ULK1 level and activity following prolonged glucose starvation. Along with the

reductions, cells lacking AMPK or reconstituted with the AA mutant ULK1 exhibited significant impairments in their ability to increase autophagy flux in response to amino acid starvation after the prolonged glucose starvation (Fig. 7l–o and Supplementary Fig. 7o, p). These results suggest that AMPK-mediated phosphorylations of ULK1 Ser556 and Thr660 play crucial roles in preserving the cellular capacity to perform autophagy and maintaining cellular resilience during prolonged glucose starvation.

## Discussion

Our study provides a rigorous demonstration that AMPK negatively regulates ULK1 activity and autophagy by phosphorylating ULK1. This challenges the widely accepted notion that AMPK promotes ULK1

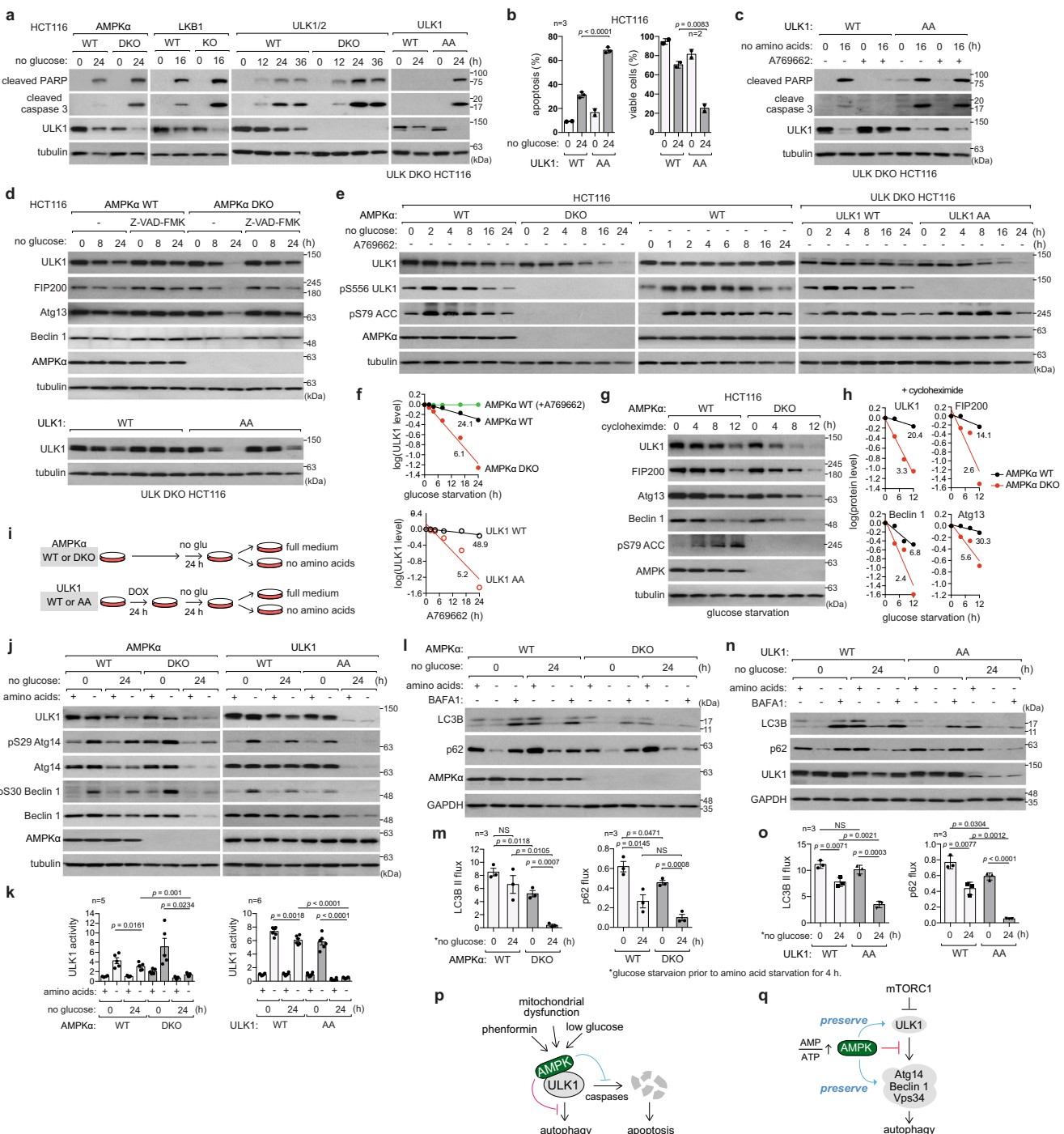

**Fig. 7 | AMPK-ULK1 axis enables cells to maintain autophagy capacity and extends cell survival during glucose starvation. a** LKB1-AMPK-ULK1 axis protects cells against apoptosis during prolonged glucose starvation. **b** Quantitation of apoptosis and viable cells. Details are described in Methods. **c** ULK1 phosphorylation at Ser556 and Thr660 protects cells against apoptosis during prolonged amino acid starvation. The WT and AA mutant HCT116 cells were analyzed. **d** AMPK activation prevents key autophagy proteins from caspase-mediated downregulation during prolonged glucose starvation, with the prevention of ULK1 downregulation being specifically mediated by Ser556/Thr660 phosphorylation. Z-VAD-FMK (25 μM) was treated during indicated hours of glucose starvation. **e** ULK1 Ser556/Thr660 phosphorylation stabilizes ULK1 during glucose starvation. A769662 was at 100 μM. **f** Quantitative analysis of proteins from (**e**) relative to WT cells/vehicle controls at the zero-time point. ULK1 half-life (hours) estimated via linear regression are indicated inside the graphs. **g** AMPK protects autophagy proteins from degradation during glucose starvation. Cycloheximide (50 μg/ml) was treated during indicated hours of glucose starvation. **h** Quantitative analysis of key

autophagy proteins from **g**. The half-life of the proteins is indicated in hours within the graphs. **i** Scheme for analyzing the recovery of autophagy following extended glucose deprivation in cells. **j** AMPK-mediated Ser556/Thr660 phosphorylations are required for efficient activation of ULK1 after prolonged glucose starvation. The indicated HCT116 cells were starved of glucose, followed by a 2-h recovery period in full or amino acid-depleted medium, as depicted in **i**. **k** Quantitation of ULK1 activity from **j**. The measurement reflects both phosphorylations of Atg14 and Beclin 1. **l–o** AMPK-mediated phosphorylations of Ser556 and Thr660 are required for the efficient autophagy flux after prolonged glucose starvation. The indicated HCT116 cells were starved of glucose, followed by 4 h amino acid starvation for autophagy flux analysis. **p**, **q** Diagram depicting two distinct functions of AMPK in regulating ULK1 and autophagy. AMPK preserves the autophagy initiation machinery (positive role depicted in blue) and suppresses ULK1 activity (negative role depicted in red). The statistical analysis in this figure was performed as in Figs. 1 and 3. Source data are provided as a Source Data file.

activity and autophagy induction. Our findings offer a mechanistic explanation for the previously reported inhibitory effects of AMPK on autophagy[1–3,10–15,17]. The key aspect of our elucidated mechanism is that AMPK-mediated phosphorylations of ULK1 at Ser556 and Thr660 suppress ULK1 activity and autophagy induction. Our study reveals that the phosphorylation of ULK1 at Ser556 by AMPK, which is broadly used as a marker of high autophagy, indeed inhibits ULK1 activation and autophagy induction. Overall, our findings redefine the role of AMPK in autophagy by demonstrating that it suppresses, rather than promotes, autophagy in energy-deprived cells.

According to the mechanism we have elucidated, AMPK regulates autophagy in two distinct ways (Fig. 7p, q). Firstly, AMPK inhibits ULK1 to prevent abrupt induction of autophagy. Upon AMPK activation, ULK1 undergoes phosphorylation, which stabilizes the AMPK-ULK1 interaction and prevents the activation of ULK1 during amino acid starvation. Secondly, AMPK protects ULK1 and other autophagy regulators associated with ULK1 from caspase-mediated degradation, preserving critical cellular components during energy stress. By preventing immediate autophagy induction and preserving essential autophagy components, AMPK enables cells to quickly resume autophagy and restore homeostasis when energy stress subsides. This protective role, together with its role in inducing autophagy gene expression (Supplementary Fig. 7d, q)[40–42], may explain why cells lacking AMPK exhibit reduced autophagy and why the murine ULK1 S555A mutation impaired mitophagy[8,9]. Our model suggests that the phosphorylation of Ser555 (human ULK1 Ser556) is not essential for ULK1 activation but rather for ULK1 stability. Cells or tissues expressing the S555A mutant for an extended period may experience reduced ability to initiate autophagy due to ULK1 destabilization.

In addition to its long-term protective role, AMPK is required for the optimal activation of the Atg14-associated Vps34 complex in response to acute amino acid starvation (Fig. 3a and Supplementary Fig. 3a, b). This role is unlikely due to the phosphorylation of Beclin 1 by AMPK since the level of Beclin 1 phosphorylation at Ser91/93 did not correlate with Vps34 activity. Notably, Vps34 activity was barely increased by glucose starvation or AMPK activation alone without amino acid starvation. This suggests that AMPK-mediated phosphorylation of Beclin 1 alone may not be sufficient for activating Vps34. We speculate that the phosphorylation might enhance the responsiveness of the Atg14-Vps34 complex to ULK1 activity for its optimal activation. Such a two-tiered regulation of the Vps34 complex by both AMPK and ULK1 may enable cells to regulate Vps34 activity specifically in the autophagy pathway without affecting its activity in other non-autophagy-related membrane processes. Similarly, the AMPK-mediated phosphorylations of Atg9, RACK1 and PAQR3[43–45] might contribute to maintaining the autophagy machinery in a state of readiness, primed for activation, rather than directly enhancing autophagy.

The restriction of autophagy during energy depletion emphasizes the tight coupling of autophagy induction with the mitochondrial activity. Our finding is consistent with the recent report that the mitochondrial oxidative chain is important for maximal autophagy[46]. Autophagosome formation occurs in the endoplasmic reticulum localized in close proximity to the mitochondria[47–49], highlighting the strong link between the mitochondrial activity and autophagy. Our study suggests that this proximity may enable efficient control of autophagy induction via the AMPK-ULK1 axis by providing ATP locally. Supporting this, the dependence of amino acid starvation-induced activation of ULK1 on the mitochondrial activity disappeared when LKB1 or AMPK was depleted. This finding indicates that the proximity of the AMPK-ULK1 axis to the mitochondria may allow for effective monitoring of the mitochondrial status and coordination of autophagy induction with the mitochondrial energetics.

The long-held notion that AMPK promotes autophagy has led to numerous studies examining the potential therapeutic applications

of AMPK-activating agents, such as metformin, for various diseases. However, conflicting results have emerged, with some studies showing adverse effects in Alzheimer's disease patients and mouse models[50–53], while others demonstrating clear benefits in ameliorating diabetes-related complications. These inconsistent outcomes may be related to the dual role of AMPK in regulating autophagy. Our findings provide a conceptual framework for reevaluating previous research and prompt the need for further investigation into the function of AMPK in different physiological and pathological contexts. Such research may deepen our understanding of how cells respond to energy stress and contribute to improving AMPK-targeting therapeutics.

## Methods
### Primary antibodies
Primary antibodies used for immunoprecipitation (IP) and western blotting (WB) were obtained and validated as described below. Unless otherwise specified, the antibodies have been validated and utilized for WB of both mouse and human proteins. When the antibodies were used for IP or immunostaining, it was explicitly noted. Antibodies for ULK1 (sc-10900 for IP and sc-33182 for WB of human protein), Atg14 (sc-164767 for IP), Beclin 1 (sc-11427 for WB and sc-10086 for IP), p62 (sc-28359), and GAPDH (sc-25778) were from Santa Cruz Biotechnology. The antibodies were validated in our previous reports[21,22,54–56]. Antibodies for ULK1 (A7481 for WB of mouse protein) and WIPI2 (SAB4200400 for immunostaining) were from Sigma-Aldrich. The antibodies were validated in our previous reports[21,22,54]. Antibodies for Vps34 (3358), Atg7 (8558), Atg9a (13509), Atg14 (rabbit monoclonal clone D3H2Z for immunostaining and WB), mTOR (2972 and 2983 for WB), phospho-Akt Ser473 (4051), Akt (9272), LC3B (2775 for WB), phospho-S6K1 Thr389 (9205), S6K1 (9202), and phospho-Atg13 Ser355 (isoform 2 Ser318; mouse Ser354) (26839) were from Cell Signaling Technology. The antibodies were validated in our previous reports[21,22,54–57]. Antibodies for AMPK (2532), LKB1 (3047), phospho-ACC (3661), phospho-ULK1 Ser758 (6888), phospho-ULK1 Ser556 (5869), phospho-ULK1 Ser317 (37762), phospho-ULK1 Ser638 (14205), cleaved PARP (9541), and cleaved caspase-3 (9661) were from Cell Signaling Technology. The antibodies were validated by the manufacturer. The validation data are available on the company's website. Myc 9E10 monoclonal antibody (OP10) and WIPI2 antibody (MABC91 for immunostaining) were from EMD-Millipore. The antibodies were validated in our previous reports[21,22,54–57]. Ubiquitin antibody (04-263) from EMD-Millipore was validated by the manufacturer. The validation data are available on the company's website. HA antibody HA.11 (901515) was from Biolegend. The antibody was validated in our previous reports[21,22,54–57]. LC3B antibody (PM036 for immunostaining) and p62 antibody (PM045 for immunostaining) were from MBL International (Woburn, MA). The antibodies were validated in our previous reports[21,22,54] for LC3B and by the manufacturer for p62. Anti-Atg13 antibodies have been described in our previous report for the validation[55]. Anti-pT660 antibody was made using PRNR(pT)LPDL-C as an antigenic peptide in rabbits, and purified using antigenic peptide-conjugated column in Abclonal Science (Woburn, MA). The antibody was validated in the current paper. Antibodies for pAtg14 Ser29 and pBeclin 1 Ser30, the sites we identified, were previously described in our reports for the validation[21,22].

### Chemicals, secondary antibodies, and other reagents
The following materials were used in the experiments. Fetal bovine serum (FBS) (F0926), Earle's Balanced Salt Solution (EBSS) (2888), Bafilomycin A$_1$ (BAFA1, B1793-10UG), blasticidin S (SBR00022-1ML), doxycycline hyclate (D9891-1G), polybrene (107689), oligomycin A (75351), antimycin A (A8674), CCCP (C2759), insulin (I0516), and dexamethasone (D2915) from Sigma-Aldrich; A769662 (A3963-50), rotenone (B5462), MG132 (A2585), MRT-68921 (B6174-5), SAR405 (A8883-2) and Z-VAD-FMK (A1902-1) from ApexBio (Houston, TX); 991 (S8654)

and GSK621 (S7898) from Selleckchem (Houston, TX); MK8722 (HY-111363), PEG300 (HY-Y0873), PEG400 (HY-Y0873A), and Tween 80 (HY-Y1891) from MedChem Express (Monmouth Junction, NJ); Hydroxychloroquine (HCQ, 26301-0250) from Acros Organics (Geel, Belgium); William's medium E (W4128), active recombinant AMPKα1/β1/γ1 complex (14-840) from EMD-Millipore; Hanks' balanced salt solution (HBSS, 14175095), penicillin and streptomycin solution (15140122), Lipofectamine 3000 (L3000015), puromycin (A11138-03), hygromycin B (10687010), anti-rabbit IgG (H + L) (A16029), anti-mouse IgG (H + L) (A16078), anti-goat IgG (H + L) (A16005), Alexa Fluor 488-conjugated anti-rabbit IgG (A-11034 and A-21441), Alexa Fluor 555-conjugated anti-mouse IgG (A-31570), and TRIzol Reagent (15-596-018) from ThermoFisher Scientific; Torin1 (4247) and rapamycin (1292/1) from R&D; phosphatidylinositol (840042 P) from Avanti Polar Lipids; PI3P Grip (G-0302) from Echelon Biosciences; Protein G-agarose bead (P9202) from GenDEPOT; polyvinylidene difluoride membrane from Bio-Rad (1620177); Dulbecco's Modified Eagle's Medium (DMEM, 25-500) and horseradish peroxidase substrate ProSignal Pico kit from Genesee Scientific (20-300B); Percoll (45001747) from Cytiva (Marlborough, MA). Recombinant Atg14 was obtained from *Escherichia coli* as a custom order to MyBioSource (San Diego, CA). All the primers used for our study were obtained from Integrated DNA Technologies (Coralville, IA) via custom synthesis.

## Cell lines and culture conditions

HCT116 (CCL-247), HeLa (CCL-2), HEK293T (CRL-11268), HepG2 (HB-8065), C2C12 (CRL-1772), and A549 cells (CCL-185) were obtained from ATCC. HT22 cells (SCC129) were obtained from EMD-Millipore. 293FT cells were obtained from ThermoFisher Scientific. All the cells used for the experiments were cultured in DMEM supplemented with 10% fetal bovine serum, penicillin, and streptomycin at 37°C in 5% $CO_2$. All cell lines were confirmed to be mycoplasma-free using MycoAlert PLUS Mycoplasma Detection kit (Lonza Walkersville, Inc. LT07). The original cell lines used in this study were authenticated by the suppliers, based on analysis of genome sequences such as short tandem repeats, isoenzyme analysis, DNA fingerprinting, and/or karyotyping. The knockout cell lines and stably transduced cells, which we generated, were validated for every experiment, in which they were used, by genotyping, western blotting, and/or DNA sequencing.

## Preparation of mouse primary hepatocytes

Primary hepatocytes were isolated from 8- to 10-week-old C57BL/6 mice using two-step collagenase perfusion method. Liver was perfused via inferior vena cava with pre-warmed (37 °C) HBSS until the liver turned pale. The liver was then perfused with 1 mg/ml Collagenase II (Worthington Biochemical Corporation, NC9693955) to digest the liver. Dissected liver was gently ruptured with forceps in HBSS and filtered through 70 μm cell strainer (22-363-548, Fisher Scientific). Released cells were centrifuged at 100×*g* for 2 min at 4 °C. Cell pellet was gently resuspended with DMEM and viable primary hepatocytes were purified using Percoll gradient centrifugation (200×*g* for 10 min at 4 °C). The cell pellet was washed and resuspended in William's E medium, then plated on collagen-coated culture plates. The culture medium was replaced after 2 h of incubation to remove non-adherent cells. Cells were cultured in William's E medium supplemented with 5% FBS, 100 nM insulin, 100 nM dexamethasone, penicillin, and streptomycin.

## Plasmid constructions and mutagenesis

Myc-tagged constructs for ULK1 and Atg13 were made using pRK5 vector as described in our previous reports[21,22,55]. For lentiviral constructs, we used pLV-EF1a-IRES plasmids (Addgene #85132, #85133, #85134). Human and mouse ULK1 genes were described in our previous report[55]. LKB1 WT and KD cDNAs were obtained from Addgene (#8590 and #8591) and subcloned into pLV-EF1a-IRES-puro. Human

and mouse ULK1 and Atg13 point mutant constructs listed in Supplementary Table 1 were generated using a site-directed mutagenesis kit (Agilent #200521). ULK1 M92 A mutant (KI mutant)[55] and LIR ULK1 mutant[54] were described previously. Mouse 4SA ULK1 mutant was obtained from Addgene (#27631). Mouse S757A and S757C mutants were kindly provided by Dr. Kun-Liang Guan (UC San Diego). The mutant constructs were cloned into pLV-EF1a-IRES-Puro with no tag. The primer sequences used for the mutagenesis are listed in Supplementary Table 1. For inducible expression, human and mouse ULK1 genes were subcloned into Lenti-TRE3G-2-PGK-Tet3G-puro (TransOMIC Technologies). Human WT AMPKα1 (Addgene #79010) and KD AMPKα1 (Addgene #79011) DNAs and rat WT AMPKα2 (Addgene #15991) and K45R AMPKα2 (kinase inactive mutant) (Addgene #15992) DNAs were subcloned into pLV-EF1a-IRES-Puro or pLV-EF1a-IRES-blast. All the generated constructs were confirmed by sequencing the DNAs at GENEWIZ (South Plainfield, NJ).

## Genome editing to generate KO cells

AMPKα DKO HCT116 cells, AMPKα DKO HeLa cells, and ULK DKO, ULK1 KO, Atg13 KO HEK293T cells were generated using the CRISPR-cas9 assisted genome editing technique. We used pSpCas9(BB)-2A-GFP (Addgene, PX458; deposited by Dr. Feng Zhang). The detailed procedures are described in our recent reports[21,22,54]. In brief, target cells were transduced by the plasmid using the Neon Transfection System (ThermoFisher Scientific, MPK5000). Two days post-transfection, green fluorescence-positive single cells were sorted and plated into 96-well plates using the Hana Single Cell Dispenser (Namocell, Mountain View, CA). Single colonies were expanded, and genomic DNA and cell extract were obtained for screening by genotyping and WB. Atg7 KO, Atg14 KO, and LKB1 KO HCT116 cells were generated using lentiCRISPRv2 vector (Addgene #98290). Lentivirus preparation and infection procedures have been described in our previous reports[55]. Briefly, lentivirus was prepared by transfecting 293FT cells (ThermoFisher Scientific, R70007) with the lentiviral vector, pHR'8.2DR and pCMV-VSV-G using Lipofectamine 3000 (ThermoFisher Scientific, L3000015). The transduced cells were cultured in the absence of antibiotics for two days. Culture supernatant that contains virus was collected 48 h after transfection and filtrated using 0.45 μm syringe filters (Genesee Scientific, 25-245). Target cells (HeLa, MEFs, and HCT116) were infected with the collected virus in the presence of polybrene at 8 μg/ml (Sigma, 107689). Stably transduced cells were selected with antibiotics. ULK DKO HCT116 cells[22], Beclin 1 KO HCT116 cells[21] and ULK DKO MEFs[21] have been described previously. AMPK DKO MEFs were obtained from Dr. Benoit Viollet. Atg9a KO MEFs were obtained from Dr. Shizuo Akira. LKB1 KO MEFs were obtained from Dr. N. Bardeesy. All the generated cell lines were verified by genotying using the primers listed in Supplementary Information and by sequencing the DNAs at GENEWIZ (South Plainfield, NJ). The sequences for gRNAs are listed in Supplementary Table 2.

## Starvation and treatment

For amino acid starvation, we used EBSS (E2888 from Sigma or SH3002902 from GE Healthcare Life Sciences) supplemented with 10% dialyzed fetal bovine serum (dFBS) (ThermoFisher Scientific, 26400044). Cells were pre-treated with chemical agents 30 minutes before they were incubated with amino acid starvation medium for 1 h to analyze ULK1 activity or interaction. For glucose starvation, we used DMEM (ThermoFisher Scientific, 11966025) supplemented with 10% dFBS. Cells were incubated with glucose starvation medium for 2 h or indicated periods of time before cells were treated with chemicals or mTORC1 inhibiting conditions. All the chemicals used were resolved in dimethyl sulfoxide (DMSO) as stock and used at the indicated concentrations for each experiment. Mitochondrial inhibitors were treated to cells in full medium at the indicated concentrations for 30 min before cells were starved of amino acids otherwise specified. All the

chemicals for analysis of their effects on ULK1 activity, Vps34 activity and autophagy were present not only in pre-incubation but also during starvation or mTORC1 inhibiting treatments.

### DNA transfection

For transient expression, cells were transfected with recombinant DNA constructs using JetOPTIMUS transfection reagent (Genesee, 55-252) following the manufacturer's protocol. HEK293T cells as target cells were seeded and cultured until the confluence reached 50–60% for transfection. Cells were harvested 2 days post-transfection for co-IP assays. For the interaction analysis, we used 200 ng of myc-tagged ULK1 DNA for transfection.

### Lentiviral preparation, viral infection, and stable cell-line generation

We used pLV-EF1a-IRES-lentiviral vectors described above to generate cell lines stably transduced with exogenous DNA. The procedures for lentivirus preparation and target cell infection have been described in our previous reports[55]. Briefly, lentivirus was prepared by transfecting 293FT cells (ThermoFisher Scientific, R70007) with pLV-EF1a-IRES-Puro construct containing a gene of interest (Addgene #85132) together with pHR'8.2DR and pCMV-VSV-G using Lipofectamine 3000 (ThermoFisher Scientific, L3000015) at 1:1:1 ratio. Viruses were collected 48-60 h after transfection, and target cells (HeLa, MEFs, and HCT116) were infected with the collected viruses in the presence of polybrene (Sigma-Aldrich, 107689). Stably transduced cells were selected with antibiotics. Because there is a limitation of DNA insert for the efficient lentiviral packaging, we could not produce pLV-EF1a-IRES vectors containing the ULK1 gene. As an alternative method, we linearized the lentiviral vector using the restriction enzyme SgrDI (ThermoFisher Scientific, ER2031) for a single cut, then introduced the linearized DNA into target cells using the Neon instrument (ThermoFisher Scientific, MPK5000). Two days after the transfection, cells harboring the lentiviral vector were selected in the presence of antibiotics.

### Co-immunoprecipitation and western blotting

To obtain whole-cell extracts for co-IP and western blot analysis, cells were lysed using a buffer containing 40 mM HEPES, pH 7.4, 120 mM NaCl, 1 mM EDTA, 50 mM NaF, 1.5 mM $Na_3VO_4$, 10 mM β-glycerophosphate, 0.3% CHAPS (Sigma, C5849), and EDTA-free protease inhibitors (Roche, 11873580001). Supernatant of cell extract obtained by centrifugation at 15,700×$g$ for 10 min at 4 °C was used for western blot analysis. To conduct co-IP, the supernatant was incubated with 2–4 μg of antibody and protein G-agarose bead (GenDEPOT, P9202) for 4 h at 4 °C. Immunoprecipitated proteins were washed 4 times using the lysis buffer, suspended in sodium dodecyl sulfate (SDS) sample buffer (50 mM Tris-HCl, pH 6.8, 2% SDS, 10% glycerol, 1% β-mercaptoethanol, 12.5 mM EDTA, 0.02% bromophenol blue), boiled for 5 min at 95 °C, then loaded onto 8% or 4-12% SDS-PAGE Tris-glycine gels (ThermoFisher Scientific, XP00085 and XP04125), and transferred onto immunoblot polyvinylidene difluoride (PVDF) membrane (Bio-Rad, 620177) using Trans-blot Turbo Transfer System (Bio-Rad, 1704150) or Criterion Blotter (Bio-Rad, 1704070). Membranes were washed briefly with phosphate buffered saline (PBS, Fisher Scientific, BP665-1) containing 0.05% Tween 20 and incubated with 5% skim milk in PBST for 1 h at room temperature. Primary antibody binding was performing using antibodies at 1:1000 dilution following the manufacture's recommendation for each antibody at 4 °C overnight. Membranes were washed four times each 10 min, then incubated with secondary antibodies listed in the key resources section. Membranes were then incubated with Enhanced Chemiluminescence WB detection reagents (Advansta, E-1119-50) to visualize protein bands. WB images were acquired by X-ray film developer or using iBright 1500 (ThermoFisher Scientific). Band intensities of WB images were quantified using Image J

(version 1.51). The half-life of protein downregulation was obtained by the Prism v6 software (version 6.0d, GraphPad Software) using the linear regression analysis and $Log_{10}(0.5)$/(slope of the best fit linear equation from the linear regression analysis).

### In vivo test of AMPK inhibition of ULK1 and autophagy

C57BL6J male mice (000664) were purchased from the Jackson Laboratories. The mice were maintained in a Specific Pathogen-Free facility with a 12-h light/dark cycle at 21-23 degrees Celsius of temperature and a humidity range of 40–60%. The mice were provided with a chow diet (2018 Harlan Teklad global 18% protein rodent diets). Mice between 10 and 12 weeks of age were used for the experiment. MK8722 was prepared in a solution containing 10% DMSO, 40% PEG300, 5% Tween 80, and 45% saline following the protocol reported previously[58]. Torin1 was prepared in a solution containing 20% N-methyl-2-pyrrolidone, 40% PEG400, and 40% water following the protocol reported previously[59]. Mice were intraperitoneally injected with MK8722 at 60 mg/kg. Fifteen minutes after MK8722 injection, Torin1 was intraperitoneally injected into the mice at 20 mg/kg. One hour after the injection of Torin1, the mouse liver and skeletal muscle were isolated on the ice and immediately frozen in liquid nitrogen. The tissue extract was prepared in a buffer containing 150 mM NaCl, 5 mM EDTA, 50 mM Tris (pH 8.0), 1% Triton X-100, 0.5% sodium deoxycholate, and 0.1% SDS. We also obtained cryosections from the frozen tissue using a cryostat (Reichert-Jung Cryocut 1800). The tissue sections were stained using anti-LC3B antibody (Cell Signaling Technology, 2775) at dilution 1:200 overnight at 4 °C. After the primary antibody labeling, the tissue sections were incubated with Alexa Flour 488-conjugated anti-rabbit IgG (ThermoFisher Scientific, A-11034) at dilution 1:500 for 2 hours at RT. Images were obtained using a Deltavision PersonelDV microscope (Applied Precision Inc., Issaquah, WA). LC3B puncta per cell (liver) and per fiber (muscle) were quantified with a specific threshold using ImageJ (version 1.51). All experimental procedures were approved by the University of Minnesota, Institutional Animal Care and Use Committee.

### Immunostaining and fluorescence microscopy

Phagophore and autophagosome formation was analyzed by immunostaining endogenous WIPI2 (EMD-Millipore, MABC91; Sigma-Aldrich, SAB4200400; dilution 1:100) and LC3B (MBL International, PM036; dilution 1:100), respectively, as we described in our previous reports[21,22]. In brief, cells were fixed with 4% formaldehyde in PBS for 15 min at room temperature, and permeabilized with 0.3% Triton X-100 for 5 min at room temperature. Permeabilized cells were incubated in PBS containing 1% BSA for 30 min and incubated with antibodies overnight at 4 °C. After the primary antibody labeling, cells were washed with PBS and incubated with Alexa Flour 488-conjugated anti-rabbit IgG at dilution 1:500 (Invitrogen, A-21441) and/or Alexa Flour 555-conjugated anti-mouse IgG at dilution 1:500 (ThermoFisher Scientific, A-31570). Nuclei were stained with DAPI (4'-6-Diamidino-2-phenylindole; Invitrogen, D-1306) at 0.1 μg/ml. Images from stained cells were obtained using a Deltavision PersonelDV microscope and analyzed by softWoRx version 6.1.3 (GE Healthcare). WIPI2 and LC3B puncta were quantified with a specific threshold using ImageJ (version 1.51).

### Autophagy flux assay

Autophagy flux was analyzed by incubating cells in full or nutrient-deprived medium in the presence or absence of BAFA1 (200 nM). We also used Torin 1 (250 nM) instead of nutrient starvation medium to induce autophagy. The concentrations of other used chemicals were: A769662 (100 μM); phenformin (2 mM); antimycin A (10 μM); rotenone (1 μM). Whole-cell extracts were prepared as described above using the lysis buffer for WB. The expression levels of LC3B and p62 in cell lysate were analyzed by WB and quantified by densitometry. The flux was

analyzed as the difference of LC3B II or p62 levels between the presence and absence of BAFA1.

## Apoptosis assay

Cells were subjected to starvation of glucose and/or amino acids or treated with phenformin and incubated during the indicated period. Cell extract was obtained using the lysis buffer for WB and analyzed for cleaved PARP and caspase 3 by WB. Additionally, cells were stained using pSIVA-IANBD specific to apoptotic cell membrane (Abcam, ab129817). Cell images were acquired using Lionheart FX fluorescence microscope (Bio-Tek, Winooski, VT) and analyzed using Gen5 program (version 3.10). Apoptotic and viable cells were also stained using apopxin green indicator and cytocalcein violet 450 (Abcam, ab176749), respectively, following the manufacturer's protocol, and analyzed by NovoCyte Quanteon flow cytometry (Agilent Technologies, Santa Clara, CA) with 488 and 405 nm as the excitation lasers and 530/30 and 445/45 nm as the detection channels.

## Aggrephagy assay

Cells were seeded at 50% confluence. Next day, cells were treated with puromycin (5 μM) in full medium for 4 h. After the puromycin treatment, cells were washed twice with fresh medium and incubated in full medium for 4 h. During the second incubation, cells were treated with chemicals or starvation medium as indicated in each experiment. After the second incubation, cells were treated with 4% formaldehyde in PBS for 10 min at room temperature, and permeabilized with 0.3% Triton X-100 for 30 min at room temperature. Ubiquitinated protein aggregates in cells were stained using anti-poly-ubiquitin antibody (EMD-Millipore, 04-263; dilution 1:100) and/or anti-p62 antibody (MBL International, PM045; dilution 1:100). The concentrations of other used chemicals are the same as described in other assays. Images were taken using a Deltavision PersonelDV microscope as described above in Immunostaining and fluorescence microscopy section. For each analysis, 50 cells were counted across three independent experiments.

## In vitro ULK1 kinase activity assay

AMPKα DKO HEK293T cells were transiently transduced with myc-ULK1 construct. Two days post-transfection, myc-ULK1 immunoprecipitates were obtained using anti-myc antibody. The isolated immunoprecipitates were incubated with 100 ng of active recombinant AMPKα/β/γ complex (EMD-Millipore, 14-840) for 30 min at RT in a reaction buffer containing 25 mM MOPS, 25 mM MgCl₂, 12.5 mM β-glycerophosphate, 5 mM EGTA, 2 mM EDTA, pH 7.2, 0.25 mM DTT, 200 μM ATP, and 100 μM AMP. As a control, the immunoprecipitates were incubated just with the buffer without AMPK. The immunoprecipitates were washed twice with a reaction buffer containing 25 mM MOPS, pH 7.5, 1 mM EGTA, 0.1 mM Na₃VO₄, 15 mM MgCl₂, and 100 μM ATP, then incubated with 50 ng of purified recombinant Atg14 for 30 min at RT in the presence or absence of ULK1 inhibitor (MRT68921) at 100 nM. The phosphorylation state of Atg14 Ser29 was analyzed by WB.

## In vitro Vps34 kinase activity assay

The lipid kinase activity of Atg14-associated Vps34 was assayed as we have described previously[21,22]. In brief, the Atg14-containing Vps34 complex was obtained by IP using anti-Atg14 antibody in a buffer containing 50 mM Tris, pH 7.4, 7.5% glycerol, 150 mM NaCl, 1 mM EDTA, and protease inhibitors (Roche, 11873580001). The immunoprecipitates were washed three times with the IP buffer and further washed once in a buffer containing 75 mM Tris, pH 7.5, 125 mM NaCl, and 12.5 mM MnCl₂. The immunoprecipitates were resuspended and incubated in a buffer containing 25 mM Tris, pH 7.5, 50 mM NaCl, 5 mM MnCl₂, and phosphatidylinositol (250 μg/ml) on ice for 10 min, which were followed by 5 additional min of incubation at RT. The Vps34 kinase reactions were initiated by adding ATP (final conc. 10 μM) and incubated for 30 min at RT. All the kinase reactions were spotted onto

nitrocellulose membrane. The membrane was blocked with 1% fat milk in PBS for 1 h, and incubated with 0.5 μg/ml of PI3P Grip (Echelon Biosciences, G-0302) in PBS containing 3% BSA (EMD-Millipore, 2960-500 G) and 0.1% Tween 20 (Sigma, 274348-4 L) for 2 h. After multiple times of extensive washing, the amount of PI3P Grip remaining on the nitrocellulose membrane was analyzed by WB using anti-GST antibody (Cytiva, 27457701). P3P dot blot intensities were quantitatively analyzed by ImageJ software (version 1.51).

## Validation of ULK1 Thr660 as a direct phosphorylation site by AMPK

HEK293T cells deficient of ULK1 were transiently transduced to express myc-tagged ULK1 constructs. Anti-myc immunoprecipitates were obtained using anti-myc antibody, then incubated with active recombinant AMPKα1/β1/γ1 complex at 100 ng (EMD Millipore 14-840) for 1 hour at RT in a reaction buffer containing 25 mM MOPS, 25 mM MgCl₂, 12.5 mM β-glycerophosphate, 5 mM EGTA, 2 mM EDTA, pH 7.2, 0.25 mM DTT, 200 μM ATP, and 100 μM AMP. The reaction was stopped by adding SDS sample buffer and analyzed by SDS-PAGE and WB. The phosphorylation states of ULK1 Ser556 and Thr660 in the reaction mixture were analyzed by WB.

## qPCR analysis of autophagy gene expression

RNAs were prepared from cells using TRIzol reagent (ThermoFisher Scientific, 15596018). We used Maxima H Minus cDNA synthesis Master Mix (ThermoFisher Scientific, TERM1662) to generate cDNAs from RNAs and SYBR Green Master Mix (ThermoFisher Scientific, A46113) for qPCR reaction following the manufacturer's protocols. Total cDNA of 10 ng was used for each reaction in 96-well plate. We used actin β (ACTB) and TATA Box Protein (TBP) genes as controls. Samples were run on QuantStudio 3 Real-Time PCR system (ThermoFisher Scientific). Results were analyzed using Microsoft Excel, version 16.54. The primers used for qPCR are listed in Supplementary Table 3.

## Statistics and reproducibility

The quantified outcomes were summarized as mean and SEM as specified in the figure legends. To compare the means between different groups, the two-tailed Student t test was used with Prism 6 (Version 6.0d, GraphPad Software). All the measurements used for statistics were obtained from distinct samples that were prepared independently of each other. The sample sizes for independent experiments and animal studies were determined based on preliminary data. The western blotting experiments were performed multiple times, as illustrated in the accompanying graphs. In cases where only micrographs were presented, the experiments were independently repeated more than three times, yielding consistent results. Representative micrographs are provided. Statistical significance was interpreted for p values below 0.05.

## Reporting summary

Further information on research design is available in the Nature Portfolio Reporting Summary linked to this article.

## Data availability

All the data that support the conclusions in this paper are available within this article and its supplementary Information file. Source data are provided with this paper as an excel file, which contains all the uncropped western blots, raw data, and quantified values for all the graphs presented in this paper.

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

## Acknowledgements

We thank M. Kundu for ULK DKO MEFs; B. Viollet for AMPK KO MEFs; N. Bardeesy for LKB1 KO MEFs; S. Akira and T. Saitoh for Atg9a KO MEFs. E. Toso and M. Kyba for cell sorting. D.H.K has received research funding from the NIH (R01GM097057 and R35GM130353).

## Author contributions

J.M.P and D.H.K conceived the project, designed and performed experiments, and wrote the manuscript. D.H.L. designed and performed experiments.

## Competing interests

The authors declare no competing interests.
