## [Peer Review File · Nature Communications]

Redefining the role of AMPK in autophagy and the energy stress responseREVIEWER COMMENTS

Reviewer #1 (Remarks to the Author):

Ji-Man Park et al. show that AMPK inhibits ULK1 to limit the phosphatidylinositol-3-phosphate, which is essential for the development of phagophores and autophagosomes. They also identified two key phosphorylations of ULK1 by AMPK that are required for the inhibition that play a role on inhibiting of ULK1 and autophagy induction. The finding was different with the “prevailing model” that AMPK activates autophagy via stimulating ULK1, which is interesting. However, the logic of this manuscript is a little confused and there are too many parts to be improved.

1. It demonstrated that AMPK activation suppresses ULK1 activity in Figure 1c, it would be helpful to detect p-ULK1.
2. At the moment, the method for observing autophagic flow is the LC3 double fluorescent labeling, in which a fluorescent protein with RFP and GFP in cells simultaneously fuses with LC3 protein, allowing the transition from the autophagosome to the autophagic lysosome to be monitored in real-time. It should complement the experiment of the RFP-GFP-LC3 assay to detect the effect of phenformin and mitochondrial inhibitors on autophagy flux (Figure 3h-3i).
3. why is not colored in figure 3j?
4. Every image of Figure 3k should be marked with a ruler.
5. MK8722 inhibited Torin1-induced LC3B puncta formation, but not the two bands of LC3B in the skeletal muscle (Figure 3m).
6. There was no internal reference in Figure 1a, 1c, 1e, 1g, 1j; Figure 2i; Figure 5a, 5b, 5d and 5f.
7. The results showed the IP findings in the absence of both negative control and positive control groups.
8. The layout of Figure 4b needs to be adjusted. There was a small distance between the bands of western blotting.
9. Does the AMPK-ULK1 interaction was detected by AMPK activator of GSK621 in Figure 4e? GSK621 stabilized the interaction but not as effectively as MK877.
10. There are many grammatical errors in the article.
11. Full-text logic is poor and lacks the main line.
12. Please check whether the wb experiment was contained the reference protein, like tubulin in figure 1

13. Please check corresponds to the analysis in this article. In figure 1, please add the band of P-S556 ULK1 and P-S758 ULK1.
14. Please check paper text again for spelling mistakes.
15. According to your description, we suggest that the author can add an experiment about the treatment of A769662 or Torin 1 or their combination in HELA,293t and HT22 cells in extended data Figurer 1.
16. The tendency of the ps29-ATG14 in Figure 1D was not correct with the description in text. Please check it.
17. Please check whether the Scale bar was added in the image in figures.
18. There are some unclearly band in the western blotting, which was unclearly identified the tendency of the expression of some protein, like the ULK1 activity towards the radial of the ps29-ATG14/ATG14, especially in the figure 2, we suggest that the author can add the quantification of the related protein in WB.
19. Please check whether the formation of the WB figure was showing no difference, like figure3K and figure3I.
20. "It is worthy of note that AMPK activation drastically suppressed S6K1 Thr389 phosphorylation but only marginally ULK1 Ser758 phosphorylation (Fig. 1a, 1e and 2j and Extended Data Fig. 2h-2j)." Please check whether the phosphorylation sites texted in the paper was consistent in the figure labeled.
21. We suggest that the author can sure the phosphorylation sites of ULK1 clearly by mass spectrometry.

Reviewer #2 (Remarks to the Author):

Park et al present a very interesting study on the molecular mechanisms that integrate nutrient starvation signals to coordinate AMPK and MTORC1 pathways to the regulation of ULK1 and autophagy. Here, these authors provide clear evidence to support the controversial view that energy stress signals activate AMPK to inhibit autophagy initiation. The current manuscript delineates the mechanism further by identifying a specific combination of AMPK-regulated sites (ULK1 Ser556 +Thr660) that play key functional roles for autophagy degradation and related anti-apoptotic pathways. In addition, a novel model is proposed in which AMPK serves to maintain global autophagy capacity by preventing destruction of regulatory complexes during energy stress, to keep the autophagy pathway primed for the period after the recovery of bioenergetic level. Overall, this reviewer found the large body of work to be well-developed, comprehensive and of high-quality. Throughout the entire collection of main figures and extended data, trends were generally internally consistent and supportive of the conclusions. There are some additional issues that should be addressed.

1. The phospho-immunoblotting are overall clear and well-controlled. One obvious deficiency is the lack of total signal to accompany phospho-ACC. The authors should address by providing this control signal, at least for several of their experiments eg the initial key signalling panels that establish the effects of nutrient starvation (Fig 1a), AMPK activators (Fig 1e), mitochondrial targeting agents (Fig 2j). This is important to definitively establish that these manipulations do not have other effects eg on total ACC in their cell systems.

2. Figure 1e. The phospho-quantification of the ULK1 substrates should not be summed and averaged together (Fig 1f and 1h). It would be preferable to plot each separately (eg pS29 Atg14/Atg14 etc.) To complement the nice analysis of ULK1-S555 in Fig 1f, these authors should also show quantification of ULK1-S758 under the same conditions.

3. In some places (eg Fig 1g,h) the separate data points are not shown in the histogram so it is unclear the number of replicates represented in the graph. For consistency, these authors should provide further details on number of biological or technical replicates represented by the data. This is more crucial for the biochemistry immunoblots. Eg none of these details can be found in legends of Fig 2-7. How many animals were analysed for Fig 3j-m? In many of the imaging quantitative analyses, n=30 cells are indicated. Are these 30 cells from the same sample and is this supported by data from independent experiments?

4. The effects of AMPK/A769662 in suppressing WIPI2 puncta and LC3 (+) autophagosome formation are clear. The accompanying immunoblots on autophagy p62 flux were less clear, since the comparison is between the BafA1 dependent effects, as comparing stimulations with AMPK activation. Eg the data establishing the baseline effect in Fig 3d clear for the p62 degradation, but the effect on LC3-II accumulation (Fig 3d lanes 3 vs 6) are less clear. In general, while all other quantification plots across the manuscript (confocal imaging and phospho-protein/total) are supported by the primary data, the flux plots are more questionable since they are showing ratios of Baf-A dependent difference. For Fig 3h, the effects of the mitochondrial inhibitors on LC3-II are even harder to see. This lack of transparency is one notable weakness in the report. It may be helpful to provide both sets of quantification, eg as extended data to show LC3-II/GAPDH for all lanes to accompany the more derivative "flux" plots.

5. Fig 3j,k: better primary image data should be provided to illustrate the changes in LC3B-puncta.

6. Fig 4d: Please clarify if the baseline cell is AMPK null or null for AMPK and ULK1 together.

7. Fig 4i is useful but it would be further helpful if this could be accompanied by a neighbouring table (eg to the right) to clarify which Figure / Extended data panel shows effects of any particular mutant.

Minor issues:

1. Figure 1b and throughout Fig 4: clarify on figure labels when Myc-IP is used.
2. Fig 2c: typo in label
3. I can suggest avoiding using AA to represent amino acid (Fig 4b, Fig 7c, Ext Fig 7d) since AA signifies the double mutant in most other cases.
4. Ext Fig 3v: phenformin label is incorrect.
5. Text: page 8. Entire sentence "ULK1 mutants harboring S638A, S638D, S758D, or S638D/S758D behaved as similarly as WT ULK1 in their interaction with AMPK in response to amino acid starvation and A769662 (Extended Data Fig. 4d, e) ... under amino acid starvation or mTORC1 inhibition." is not clear and difficult to decipher.
6. Text: page 9. "These results suggest that AMPK phosphorylates Ser556 and Thr660 to protect Ser758 against dephosphorylation thereby stabilizing the AMPK-ULK1 interaction and enhancing ULK1 activity (Fig. 5h)" Do the authors mean an inhibition of ULK1 activity? To further clarify, it would help to add onto Fig 5h the status of AMPK activation and placement of S758 dephosphorylation into the schematic.
7. Page 11. The section/paragraph describing AMPK and Caspase dependent regulation of ULK1 levels (Fig 7e and Ext Fig 7f-p) is particularly dense and hard to follow. Overall, the narrative here should be improved for clarity. The plots in Fig 7e are less clear than the simplified format in Ext Fig 7n and 7p. The strong effect of A769662 treatment on ULK1 stabilisation (more pronounced than glucose starvation) is not mentioned.

Reviewer #1 (Remarks to the Author):

Ji-Man Park et al. show that AMPK inhibits ULK1 to limit the phosphatidylinositol-3-phosphate, which is essential for the development of phagophores and autophagosomes. They also identified two key phosphorylations of ULK1 by AMPK that are required for the inhibition that play a role on inhibiting of ULK1 and autophagy induction. The finding was different with the “prevailing model” that AMPK activates autophagy via stimulating ULK1, which is interesting. However, the logic of this manuscript is a little confused and there are too many parts to be improved.

1. It demonstrated that AMPK activation suppresses ULK1 activity in Figure 1c, it would be helpful to detect p-ULK1.

Thanks for the comment. We have now added the p-ULK1 blot in Figure 1c.

2. At the moment, the method for observing autophagic flow is the LC3 double fluorescent labeling, in which a fluorescent protein with RFP and GFP in cells simultaneously fuses with LC3 protein, allowing the transition from the autophagosome to the autophagic lysosome to be monitored in real-time. It should complement the experiment of the RFP-GFP-LC3 assay to detect the effect of phenformin and mitochondrial inhibitors on autophagy flux (Figure 3h-3i).

We appreciate the reviewer’s comment. While the RFP-GFP-LC3B assay is widely used in the field to monitor autophagy, we did not utilize it in our study due to concerns about potential interference with autophagy initiation by overexpressed LC3B (PMID: 31208283). As our study focuses specifically on the mechanism of autophagy initiation, we deemed it important to avoid potential confounding factors. Although the dual tag marker is useful for monitoring autophagosome-lysosome fusion in the later stages of autophagy, it is not appropriate for our study. We hope that this explanation clarifies our reasoning.

To avoid potential artifacts, we have chosen to analyze endogenous molecules rather than rely on overexpression markers, which can disturb the integrity of the autophagy pathway and alter the stoichiometry of the complexes. We would like to emphasize that we have rigorously conducted our autophagy and autophagy flux assays using 5-6 different methods, all of which adhere to the guidelines for monitoring autophagy (PMID: 33634751). We’d appreciate that the reviewer recognizes the thoroughness of our approach.

3. why is not colored in figure 3j?

As the reviewer may be aware, microscope signals are captured as digital inputs that can be displayed in various colors on a screen. Our understanding is that black and white images can provide better contrast and resolution for small cellular structures compared to other color schemes. However, we acknowledge that the choice of color scheme used to present our results does not impact the quality of our findings. In response to the reviewer’s concern, we have converted the image to a colored format. We appreciate the reviewer’s input.

4. Every image of Figure 3k should be marked with a ruler.

We have addressed the reviewer’s suggestion by adding rulers to the main images in Figure 3k.

5. MK8722 inhibited Torin1-induced LC3B puncta formation, but not the two bands of LC3B in the skeletal muscle (Figure 3m).

We apologize for any confusion and appreciate the reviewer’s comment. We interpret the comment as saying that the two bands of LC3B in the muscle sample shown in the western blot data are ambiguous. To address this concern, we have replaced the blot with a new one that provides clearer resolution of the bands. We also quantified the LC3B II band per GAPDH to make clear the effect of MK8722 on LC3B. We hope this addresses the reviewer’s concern.

6. There was no internal reference in Figure 1a, 1c, 1e, 1g, 1j; Figure 2i; Figure 5a, 5b, 5d and 5f.

Thanks for bringing this concern to our attention. We have addressed the concern by including tubulin blots for the figures. The revised figures are as follows: Figure 1a, 1c, 1f, 1h, 1i; Figure 2j; Figure 5a, 5b, 5d and 5f.

7. The results showed the IP findings in the absence of both negative control and positive control groups.

We appreciate the reviewer's comment. To address the concern, we have included a new figure (Supplementary Figure 4a) that shows a negative control using an IgG IP control. This control demonstrates the specificity of the interaction between AMPK and ULK1. This new data also confirms that ULK1 interacts with Atg13, the binding partner of ULK1. In addition, we have included a control by confirming that AMPK is pulled down by Atg13, further supporting the specific interaction of AMPK with the ULK1 complex.

8. The layout of Figure 4b needs to be adjusted. There was a small distance between the bands of western blotting.

We appreciate the reviewer's comment, however, we were unsure about the meaning of "a small distance between the bands." We assumed that it might be about the arrangement of the blots in the figure. We have adjusted the blots slightly, hoping that this addresses the reviewer's concern.

9. Does the AMPK-ULK1 interaction was detected by AMPK activator of GSK621 in Figure 4e? GSK621 stabilized the interaction but not as effectively as MK877.

While GSK621 had a weaker effect than other AMPK activators, it still prevented the destabilization of the interaction by amino acid starvation. To strengthen the conclusion, we conducted an additional experiment, quantified the interaction, and added the data in Supplementary Figure 4c and d.

10. There are many grammatical errors in the article.

Thanks for bringing this to our attention. We have carefully reviewed and corrected grammatical errors present throughout the manuscript.

11. Full-text logic is poor and lacks the main line.

Thanks for the comment. We have carefully reviewed the paragraphs throughout the manuscript and have taken steps to strengthen the main point of the manuscript. We have rephrased sentences to ensure that our argument is clearer. This change has been made thoroughly over the manuscript.

12. Please check whether the wb experiment was contained the reference protein, like tubulin in figure 1.

Thanks for the comment. As we responded to the comment 6 above, we have included tubulin as a loading control in the majority of our experimental sets, and have made sure to use it as a reference protein in the blots to strengthen our presented results.

13. Please check corresponds to the analysis in this article. In figure 1, please add the band of P-S556 ULK1 and P-S758 ULK1.

To address this concern, we have included new blots for P-S556 ULK1 and P-S758 ULK1 in every set presented in Figure 1.

14. Please check paper text again for spelling mistakes.

We have corrected spelling mistakes throughout the manuscript.

15. According to your description, we suggest that the author can add an experiment about the treatment of A769662 or Torin 1 or their combination in HELA,293t and HT22 cells in extended data Figurer 1.

We performed a new experiment as suggested by the reviewer and added the result to Supplementary Figure 1g.

16. The tendency of the ps29-ATG14 in Figure 1D was not correct with the description in text. Please check it.

We have revised the part in the main text (page 3) to properly describe the result presented in the blots.

17. Please check whether the Scale bar was added in the image in figures.

We have included scale bars for every image data in the main figures and the supplementary figures.

18. There are some unclearly band in the western blotting, which was unclearly identified the tendency of the expression of some protein, like the ULK1 activity towards the radial of the ps29-ATG14/ATG14, especially in the figure 2, we suggest that the author can add the quantification of the related protein in WB.

Thanks for the reviewer's comment. To address the concern, we have quantified the ratio of P-Atg14 to Atg14 for every blot in Figure 2 that measures ULK1 activity. This analysis provides a more precise and quantitative evaluation of the presented blots, and we believe it clarifies any potential ambiguity.

19. Please check whether the formation of the WB figure was showing no difference, like figure3K and figure3I.

We appreciate the reviewer's comment. We have added quantification data for the LC3B blots and presented it in a graph format in Figure 3m. We hope this addresses the reviewer's concern.

20. "It is worthy of note that AMPK activation drastically suppressed S6K1 Thr389 phosphorylation but only marginally ULK1 Ser758 phosphorylation (Fig. 1a, 1e and 2j and Extended Data Fig. 2h-2j)." Please check whether the phosphorylation sites texted in the paper was consistent in the figure labeled.

Thanks for the comment. We have carefully checked the labeled figures and their corresponding phosphorylations, and have made the necessary revisions. Specifically, we have revised the labeling in Figures 1a, 1f, 2k and Supplementary Fig. 2h-j.

21. We suggest that the author can sure the phosphorylation sites of ULK1 clearly by mass spectrometry.

We appreciate the valuable suggestion. As mentioned in the main text (page 8) with the references provided, all the phosphorylations we describe in our paper have been previously identified by mass spectrometry. However, validating that a site is indeed an AMPK target site requires an assay that confirms endogenous phosphorylation. Using phospho-specific antibodies is the most reliable approach for this purpose.

To validate that Thr660 is an AMPK target site, we developed new phospho-specific antibodies and demonstrated endogenous phosphorylation, along with confirming that AMPK directly phosphorylates the site (Supplementary Fig. 4k, l). This has finally defined Thr660 as an AMPK target site, and we have validated that the antibodies are highly specific to the phosphorylation state of endogenous ULK1 Thr660. Our approach provides the validation for the previously identified phosphorylation site. We have revised the manuscript to make this point clear (page 8).

We would like to express our appreciation for the time and effort that the reviewer has taken to provide the constructive comments. We hope that our responses above could have sufficiently addressed all the reviewer's concerns.

Reviewer #2 (Remarks to the Author):

Park et al present a very interesting study on the molecular mechanisms that integrate nutrient starvation signals to coordinate AMPK and MTORC1 pathways to the regulation of ULK1 and autophagy. Here, these authors provide clear evidence to support the controversial view that energy stress signals activate AMPK to inhibit autophagy initiation. The current manuscript delineates the mechanism further by identifying a specific combination of AMPK-regulated sites (ULK1 Ser556 +Thr660) that play key functional roles for autophagy degradation and related anti-apoptotic pathways. In addition, a novel model is proposed in which AMPK serves to maintain global autophagy capacity by preventing destruction of regulatory complexes during energy stress, to keep the autophagy pathway primed for the period after the recovery of bioenergetic level. Overall, this reviewer found the large body of work to be well-developed, comprehensive and of high-quality. Throughout the entire collection of main figures and extended data, trends were generally internally consistent and supportive of the conclusions. There are some additional issues that should be addressed.

1. The phospho-immunoblotting are overall clear and well-controlled. One obvious deficiency is the lack of total signal to accompany phospho-ACC. The authors should address by providing this control signal, at least for several of their experiments eg the initial key signalling panels that establish the effects of nutrient starvation (Fig 1a), AMPK activators (Fig 1e), mitochondrial targeting agents (Fig 2j). This is important to definitively establish that these manipulations do not have other effects eg on total ACC in their cell systems.

Thanks for the comment. We have added total ACC blots for the figures specified by the reviewer, as well as for other relevant figures in the manuscript.

2. Figure 1e. The phospho-quantification of the ULK1 substrates should not be summed and averaged together (Fig 1f and 1h). It would be preferable to plot each separately (eg pS29 Atg14/Atg14 etc.) To complement the nice analysis of ULK1-S555 in Fig 1f, these authors should also show quantification of ULK1-S758 under the same conditions.

Thanks for the valuable comment. To address the concern, we have reorganized the presentation of our data and separately quantified the phosphorylations, which are now presented in separate graphs. Specifically, Figure 1g now shows separate graphs for pAtg14/Atg14, pULK1 p556, and pULK1 S758, respectively. In addition, we have added quantified data in Figure 1e, i, and k, where we present pAtg14/Atg14 and pBeclin1/Beclin1 separately.

3. In some places (eg Fig 1g,h) the separate data points are not shown in the histogram so it is unclear the number of replicates represented in the graph. For consistency, these authors should provide further details on number of biological or technical replicates represented by the data. This is more crucial for the biochemistry immunoblots. Eg none of these details can be found in legends of Fig 2-7. How many animals were analysed for Fig 3j-m? In many of the imaging quantitative analyses, n=30 cells are indicated. Are these 30 cells from the same sample and is this supported by data from independent experiments?

Thanks for the important point. We have included the detail of information on the replication numbers for experiments in figure legends and now every graph has data points. The cell images and tissue studies have been analyzed across more than three separate experiments. We have made clear this information in figure legends.

4. The effects of AMPK/A769662 in suppressing WIPI2 puncta and LC3 (+) autophagosome formation are clear. The accompanying immunoblots on autophagy p62 flux were less clear, since the comparison is between the BafA1 dependent effects, as comparing stimulations with AMPK activation. Eg the data establishing the baseline effect in Fig 3d clear for the p62 degradation, but the effect on LC3-II accumulation (Fig 3d lanes 3 vs 6) are less clear. In general, while all other quantification plots across the manuscript (confocal imaging and phospho-protein/total) are supported by the primary data, the flux plots are more questionable since they are showing ratios of Baf-A dependent difference. For Fig 3h, the effects of the mitochondrial inhibitors on LC3-II are even harder to see. This lack of

transparency is one notable weakness in the report. It may be helpful to provide both sets of quantification, eg as extended data to show LC3-II/GAPDH for all lanes to accompany the more derivative “flux” plots.

We appreciate the reviewer’s careful assessment of the data. We have included all the measured analysis data in supplementary figures for all the flux experiments. Regarding Figure 3d, we acknowledge that we did not anticipate a significant difference in p62 level with starvation/BafA1 in lanes 3 vs 6. Instead, the key comparison should be made between lanes 2 and 3 vs lanes 5 and 6, which reflect autophagy flux.

5. Fig 3j,k: better primary image data should be provided to illustrate the changes in LC3B-puncta.

Thanks for the comment. We have included better primary image data for the tissue sections. We have replaced the main figures and added the primary images in Supplementary Figure 3s.

6. Fig 4d: Please clarify if the baseline cell is AMPK null or null for AMPK and ULK1 together.

The baseline cells for Figure 4d were HEK293T cells where AMPK α 1 and α 2 are lacking. ULK1 is intact in the cells. We have made this clear in the figure legend.

7. Fig 4i is useful but it would be further helpful if this could be accompanied by a neighbouring table (eg to the right) to clarify which Figure / Extended data panel shows effects of any particular mutant.

Thanks for the suggestion. We agree that such a further elaboration can help. We have added the information on the right side of the cartoon.

Minor issues:

1. Figure 1b and throughout Fig 4: clarify on figure labels when Myc-IP is used.

We have clarified the labels in those figures.

2. Fig 2c: typo in label

Thanks. We have corrected the typo.

3. I can suggest avoiding using AA to represent amino acid (Fig 4b, Fig 7c, Ext Fig 7d) since AA signifies the double mutant in most other cases.

Thanks for the suggestion. We replaced “AA” with “amino acids” throughout the manuscript to distinguish it from the double mutant.

4. Ext Fig 3v: phenformin label is incorrect.

Thanks. We have corrected the typo.

5. Text: page 8. Entire sentence “ULK1 mutants harboring S638A, S638D, S758D, or S638D/S758D behaved as similarly as WT ULK1 in their interaction with AMPK in response to amino acid starvation and A769662 (Extended Data Fig. 4d, e) ... under amino acid starvation or mTORC1 inhibition.” is not clear and difficult to decipher.

We have revised the sentences. The corrected sentences are “*ULK1 mutants with S638A, S638D, S758D, or S638D/S758D mutations showed similar interaction with AMPK as WT ULK1 in response to amino acid starvation and A769662. These results suggest the existence of AMPK-dependent events, beyond mTORC1-mediated phosphorylation of Ser758, that stabilize the AMPK-ULK1 interaction.*”

6. Text: page 9. “These results suggest that AMPK phosphorylates Ser556 and Thr660 to protect Ser758 against dephosphorylation thereby stabilizing the AMPK-ULK1 interaction and enhancing ULK1

activity (Fig. 5h)” Do the authors mean an inhibition of ULK1 activity? To further clarify, it would help to add onto Fig 5h the status of AMPK activation and placement of S758 dephosphorylation into the schematic.

Thanks for bringing the mistake to our attention. We have revised the sentence. The revised sentence is “*These results suggest that AMPK-mediated phosphorylations at Ser556 and Thr660 stabilize Ser758 phosphorylation, enhancing the stability of the AMPK-ULK1 interaction and suppressing ULK1 activation (Fig. 5h).*” It is on page 9. We have revised the model diagram incorporating the reviewer’s points.

7. Page 11. The section/paragraph describing AMPK and Caspase dependent regulation of ULK1 levels (Fig 7e and Ext Fig 7f-p) is particularly dense and hard to follow. Overall, the narrative here should be improved for clarity. The plots in Fig 7e are less clear than the simplified format in Ext Fig 7n and 7p. The strong effect of A769662 treatment on ULK1 stabilisation (more pronounced than glucose starvation) is not mentioned.

Thanks for the valuable comment. We have thoroughly revised the relevant sections on page 10 and 11 to improve the clarity, and we have also simplified and clarified the plots in Figure 7 for the stability analysis. In addition, we have included a discussion on the effect of A769662 on ULK1 stabilization in the last paragraph on page 10.

We would like to sincerely appreciate the reviewer for the valuable, constructive, and thoughtful comments. We hope that our responses above could have satisfactorily addressed all the reviewer’s concerns.

REVIEWERS' COMMENTS

Reviewer #2 (Remarks to the Author):

The authors have done an excellent job addressing my set of comments. I can advise that some additional control blots were added. Further details, updates to result presentation format, and minor corrections to inconsistencies in text were appropriately added.

One further minor comment: I had asked for more clarity by re-plotting the separate values for the autophagy flux experiments in Fig 3. The authors were able to satisfy this request and added the full set of blots in Suppl Fig 3. I should ask that the authors further clarify in the text that Main Fig 3 d,e correspond specifically to Supp Fig 3g. Currently it lists Supp Fig 3f-l to correspond to Fig 3d,e (but Supp Fig 3h-l look like they use HCT116 cells), and it is not rigorous to bounce back and forth between host cell types without proper signage.

Reviewer #2 (Remarks to the Author):

The authors have done an excellent job addressing my set of comments. I can advise that some additional control blots were added. Further details, updates to result presentation format, and minor corrections to inconsistencies in text were appropriately added.

One further minor comment: I had asked for more clarity by re-plotting the separate values for the autophagy flux experiments in Fig 3. The authors were able to satisfy this request and added the full set of blots in Suppl Fig 3. I should ask that the authors further clarify in the text that Main Fig 3 d,e correspond specifically to Supp Fig 3g. Currently it lists Supp Fig 3f-l to correspond to Fig 3d,e (but Supp Fig 3h-l look like they use HCT116 cells), and it is not rigorous to bounce back and forth between host cell types without proper signage.

We appreciate the reviewer's feedback regarding the confusion in the main text. To improve clarity, we have revised the description by separating the figure annotations by the different cell types. The revised description can be found in the first paragraph of page 6.

We would like to express our sincere appreciation to the reviewer for the thorough assessment of the clarity and presentation of the data.